

# Multi-hazard Tropical Cyclone Risk Assessment for Australia

Cameron Do [1,2,3] and Yuriy Kuleshov [1,2]

1 Bureau of Meteorology, 700 Collins Street, Melbourne, Victoria 3008, Australia
2 Royal Melbourne University of Technology (RMIT) University, 124 La Trobe Street, Melbourne,
5   Victoria 3000, Australia
Monash University, Clayton Campus, Wellington Road, Victoria 3800, Melbourne, Australia

Corresponding authos: Yuriy Kuleshov yuriy.kuleshov@bom.gov.au



**Abstract**

Tropical cyclones (TCs) have long posed a significant threat to Australia's population, infrastructure, and natural environment. This threat may grow under climate change as projections indicate continuing sea level rise and increases in rainfall during TC events. Previous TC risk reduction efforts have focused on the risk from wind alone, whereas a holistic approach requires multi-hazard risk assessments that also consider impacts of other TC-related hazards. This study assessed and mapped

TC risk nationwide, focusing on the impacts on population and infrastructure from the TC-related hazards of wind, storm surge, flooding and landslides. Risk maps were created at the Local Government Area (LGA) level for all of Australia, using collated data on multiple hazards, exposure and vulnerability. The study demonstrated that the risk posed by all hazards was highest for coastal LGAs of eastern Queensland and New South Wales followed by medium risk across Northern

Territory and north-west of Western Australia, with flood and landslide hazards also affecting several inland LGAs. The resulting maps of risk will provide decision-makers with the information needed to further reduce TC risk, save lives, protect the environment, and reduce economic losses.

## 1. Introduction

Tropical Cyclones (TCs), also known as hurricanes or typhoons, are powerful and highly destructive

meteorological hazards. Since 1970, almost 2,000 natural disasters have been attributed to TCs, which has led to over 700,000 deaths worldwide (World Meteorological Organisation, 2021). Costing about U.S.$26 billion annually in global damages (Mendelsohn et al., 2012), their impact is expected to multiply to U.S.$60 billion annually by 2100 (Bakkensen and Mendelsohn, 2019).

The proportion of intense TCs (categories 4-5) and peak wind speeds of the most intense TCs are

projected to increase at the global scale with increasing global warming (high confidence) (IPCC AR6) (ICC, 2021). The potential of more destructive TC events will require updating and enhancement of existing risk reduction strategy. The Sendai Framework for Disaster Risk Reduction provides a structure for reducing disaster damages and increasing resilience to hazards including TCs (Bennett, 2020). One mechanism they encourage in Goals 18 and 24 is the distribution of multi-hazard risk

information such as risk assessments.

Risk assessments combine hazard information with human activity, infrastructure and natural resources to determine the possible impacts of hazardous events (Belluck et al., 2006; National Research Council, 1991) and make informed choices for risk management in the most exposed and vulnerable regions (Aguirre-Ayerbe et al., 2018). Disaster risk is defined as the probability of harmful

consequences, or significant losses, resulting from interactions between a hazard, and the local exposure and vulnerability to that hazard (Crichton, 1999; Downing, 2001).

As Local Government Areas (LGAs) are the smallest government decision-making body, information is sought to be provided on that scale. Risk assessments are a foundation for early warning systems to raise alerts of potential impacts, and to provide evidence for the prioritisation of funds and

resources to areas in advance of any hazardous events. While the climate continues to change alongside evolving human activity, risk assessments must likewise be regularly updated to stay accurate and useful as a tool for disaster risk reduction (Peduzzi et al., 2012).



For TCs, the four main hazards are the destructive winds, associated storm surge, flooding from associated heavy rainfall, and landslides on steep terrain as soils saturate (Murray et al., 2020). TCs and other natural hazards are becoming increasingly recognised as multi-hazardous in nature (Scawthorn et al., 2006). These hazards impact regions differently and their effects can compound to cause even greater damage (Gori et al., 2020).

While TCs can cause damage through different hazards, such as gale-force winds, storm surge or flooding, the communication of TC intensity and categorisation places emphasis on wind speed (Lavender and Mcbride, 2020). This is partially due to the availability of wind measuring technology and the relative ease to quantify wind. Publicly available warnings and forecasts are focusing on wind speeds, ultimately portraying the message that winds are the hazard to be most wary of. The literature however suggests the impacts of storm surge and flooding contribute to the most human lives lost and infrastructure damage (Mendelsohn et al., 2012; Zhang et al., 2008). Although some studies have included multi-hazard aspects of TCs (Burston et al., 2017), presenting different hazard models for storm surge, wind and flooding, these studies do not complete the story of combining hazard with exposure and vulnerability to map risk. Similarly, within the literature, there are many examples of standalone exposure or vulnerability index assessments for TCs (Marín-Monroy et al., 2020; Bathi and Das, 2016; Amadio et al., 2019). This gap indicates compelling scope to develop a multi-hazard TC risk assessment that can differentiate the extent and severity of TC-related hazards.

This study will address this gap and strengthen TC risk information for the Australian region. Multi-hazard risk is assessed and visualised through interactive maps which show LGA categorisation, alongside hazard, exposure, and vulnerability layers.

## 2. Data and Methodology

To calculate the multi-hazard risk of TCs to Australia, hazard, exposure and vulnerability datasets were chosen and sourced. This data was then joined to LGA map shapefiles in ArcGIS Pro. To calculate exposure and vulnerability indexes from multiple indicators, equal weighting was used for exposure, while Pareto front-ranking was used for vulnerability. Combined with hazard values for each LGA, exposure and vulnerability indexes were used to calculate risk using equation 1:

$$\text{Risk} = \text{Hazard} \times \text{Exposure} \times \text{Vulnerability} \tag{1}$$

### 2.1. Selection of indicators

**Hazard**

The main identified hazards of TCs include storm surge, winds, landslides, and floods. The 100 year return period was chosen to represent the danger of these hazards in the near future.

Storm surge and wind datasets were specifically designed for TCs (Cardona et al., 2014; Arthur, 2021), and spatial mean values were calculated over each LGA. For flood and landslide hazards the original datasets did not consider solely TC induced floods/landslides. Thus the flood and landslide hazards were weighted towards TC prone regions by multiplying values by the TC wind raster dataset. Weighted flood and landslide values were then summed over LGAs as there were many null values. Greater than zero values exist only around water catchments and rivers for floods, and


around mountain regions for landslides. Thus LGAs with higher flood and landslide values have more of these prone environments in total rather than a higher areal proportion.

**Exposure**

Exposure indicators of population, hospitals, substations, and power lines were chosen to represent physical assets of human life, as well as systems and infrastructure that are important in the case of emergency disaster events (hospitals, power). Failure to maintain the function of lifeline infrastructures such as hospitals and power can lead to exacerbated negative impacts (Ju et al., 2019). These chosen indicators aim to spatially describe which LGA regions have more exposed assets relative to the rest of the country. Electrical substations provide power as critical infrastructure and are strategically placed to meet demand. Similar reasoning influenced the choice of public hospitals and powerlines.

While population density data of each LGA was found in tabular form from the Australian Bureau of Statistics (ABS), the remaining exposure indicators' raw format was as point or line shapefiles displayable in ArcGIS Pro. Thus geoprocessing tools such as spatial join were used to count the number of public hospitals in each LGA. Using absolute measurements can be inappropriate when considering regions of different sizes (Rygel et al., 2006), thus these counts were then divided by LGA area to give a density value similar to that of population density.

**Vulnerability**

Vulnerability indicators were chosen to represent regions most susceptible to high impact from a TC event occurring in the vicinity. Measures of socioeconomic status are commonly used to describe vulnerability to natural hazard events (Mitsova et al., 2018; Lianxiao and Morimoto, 2019) and the Index of Relative Socioeconomic Disadvantage (IRSD) has been used in previous literature for the Australian region (Rolfe et al., 2020). It summarises variables about the social and economic conditions of households. The more disadvantaged a region is socioeconomically, the more likely it will be more impacted by TCs, due to factors such as lower income, having families with only one parent or having a higher percentage of people that have English as a second language. The 'no vehicle homes' indicator was derived by calculating the percentage of homes with no vehicles, and the 'vulnerable age groups' indicator was constructed by calculating the percentage of an LGA's population made up of the <15 and >65 age group combined. The 'no vehicle homes' indicator is particularly relevant to TCs as it provides information on LGAs that are more susceptible to loss of human life in evacuation situations.

The data that was used to create the risk maps are summarised in Appendix.

2.2. TC Risk Mapping

The TC risk mapping process is schematically described in Figure 1. Before risk could be calculated and mapped based on the collected datasets, data was transformed and converted, as described in the diagram. Most processes occurred within ArcGIS Pro software, however, Python scripts were also utilised for some calculations.


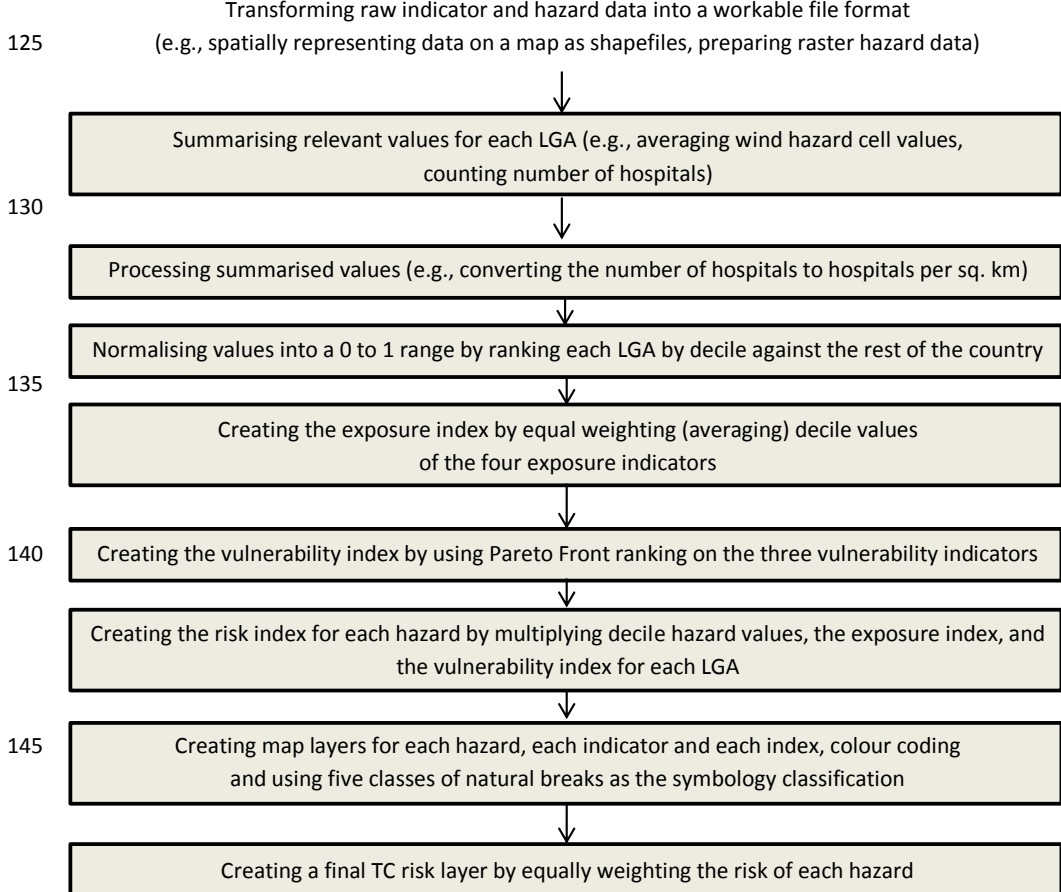






**Figure 1**.TC risk mapping process.

2.3. Indices Calculation

First, for processing raw indicator data, decile and natural breaks transformations were explored.

Decile ranking in this context compares the values of each LGA to the LGAs in the rest of the country.
A value of 0.9 would indicate the LGA has a value larger than 90% of LGAs in Australia, and every 0.1
interval would hold 10% of LGAs. In this way, all indicators can effectively have an impact on
resultant indices and risk maps even with the presence of outliers, which will take decile values on
either end of the spectrum without causing any skew. Decile ranking is used in indices such as Socio-
Economic Indexes for Areas (SEIFA) from which IRSD is a part of, to give relative meaning to the raw
scores.

Natural breaks can similarly address the limitations of 0 to 1 normalisation by using optimisation to
categorise values and minimise the amount of variance within each category. The number of
categories can be increased automatically until a threshold of variance is met (96% in our case, as
97% required more than 20 categories). The breaks or classes chosen depends on and is unique to


every distribution or set of data. Additionally, the number of classes is not fixed, which can result in
fewer unique values and less value variation between LGAs, which is less informative.

Based on these considerations, decile ranking was chosen as the method of processing raw indicator data. Natural breaks however were used in the presentation of maps and colour classes as it is the standard in geographical mapping for choropleth maps (Anchang et al., 2016), providing a quick overview and differentiating values more clearly than a continuous scale.

Second, index calculations were performed. Equal weighting is commonly used to create index values from a set of indicators, and is used either for simplicity or because there is no supporting evidence to suggest how different indicators should be weighted (Rygel et al., 2006). In the context of TCs in Australia, while past studies have suggested that a weighted framework could improve results (Do and Kuleshov, 2022), it would require more research - such as gathering expert opinion -
to weigh chosen indicators. One of the limitations of equal weighting is that very high values in one indicator are averaged with other indicators in the index, resulting in a potentially lower value that does not capture the extreme aspect of that LGA. This is particularly a problem for the vulnerability index because a region only needs to be extremely vulnerable in one factor to be considerably more at risk (Rygel et al., 2006).

Pareto ranking, also known as Pareto front optimisation or multi-objective optimisation, was investigated to address some of the limitations of the equal weighting method. Pareto ranking can be used to construct an effective vulnerability index without weighting individual indicators (Huang et al., 2013; Nelson et al., 2020). It involves finding the values along the Pareto front, which are values considered to be non-dominated in all indicator axes and ranking these fronts in order. The
process is depicted in Figure 2 which shows a step-by-step process of identifying non-dominated data points.

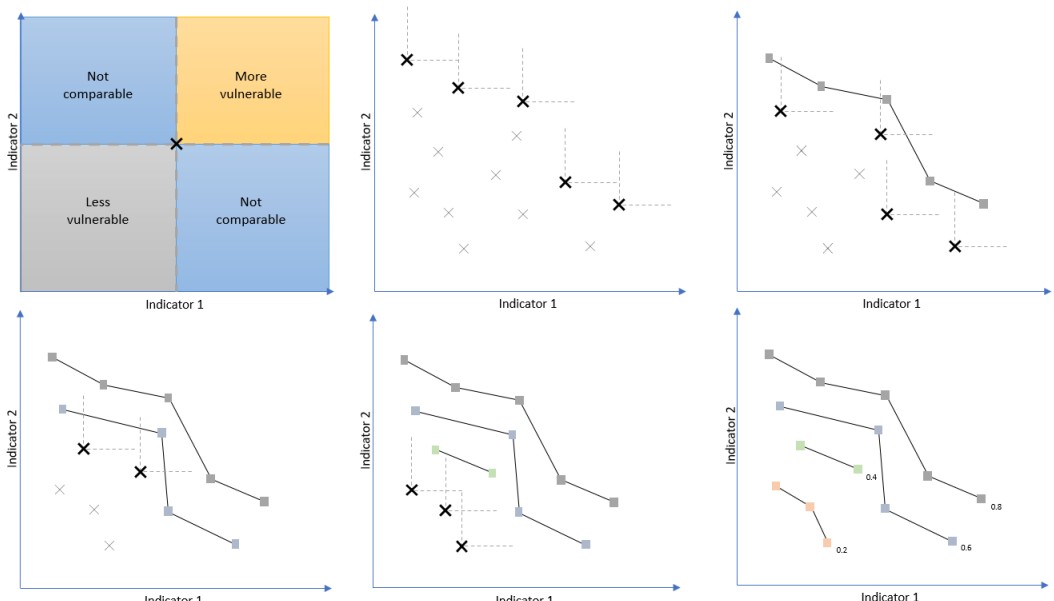

**Figure 2**. Graphic demonstration of Pareto front classification in two dimensions. The same principle applies when scaled to N number of dimensions. Adapted from Rygel et al. (2016)

First data is plotted along axes representing each component/indicator. Each data point in this study would represent an Australian LGA. Then the first non-dominated front would be identified as the set of points that do not have any LGAs with both a higher value in indicator 1 or indicator 2. This first front would be ranked highest and set aside, with the same methodology being used to identify subsequent fronts. In the case of the example in Figure 2, with 4 distinct fronts or classes, an index

value would be given at even intervals (e.g. 0.2, 0.4, 0.6, and 0.8) with LGAs sharing the same index value as LGAs also in their front.

The Pareto ranking method, therefore, can identify LGAs as vulnerable due to one or two indicator values even if its other indicator values are lower. Although vulnerability benefits from Pareto ranking as the maximum magnitude across all indicators is the defining factor, the exposure index

benefits from taking into account all indicators cumulatively assuming the selected indicators are relevant. Thus Pareto ranking was used to calculate the vulnerability index in this study, while equal weighting was chosen for the exposure index.

### 3. Results and Discussion

The hazard, exposure, vulnerability, and risk maps are presented and discussed in this section.

### 3.1. Exposure

The exposure index was created from equally weighting the four indicators: population density, hospital density, electrical substation density and powerline length density. In Figure 3, it can be seen that population density is highest along the coast and surrounding major cities, especially in New South Wales (NSW) and Victoria (VIC). The hospital density indicator shows very similar

patterns although there are fewer LGAs with the lowest exposure classification. Substation and





powerline indicators both have similar patterns to each other with the highest exposure along
south-western Western Australia (WA), southern South Australia (SA), most of VIC, and eastern NSW
and Queensland (QLD). The calculated exposure index in Figure 4 maintains the clear trends of
highest exposure along the country's eastern coast, and around major cities. Also of note are

relatively high exposure values around the Pilbara region in north-western WA and the Mount Isa
LGA in western QLD.

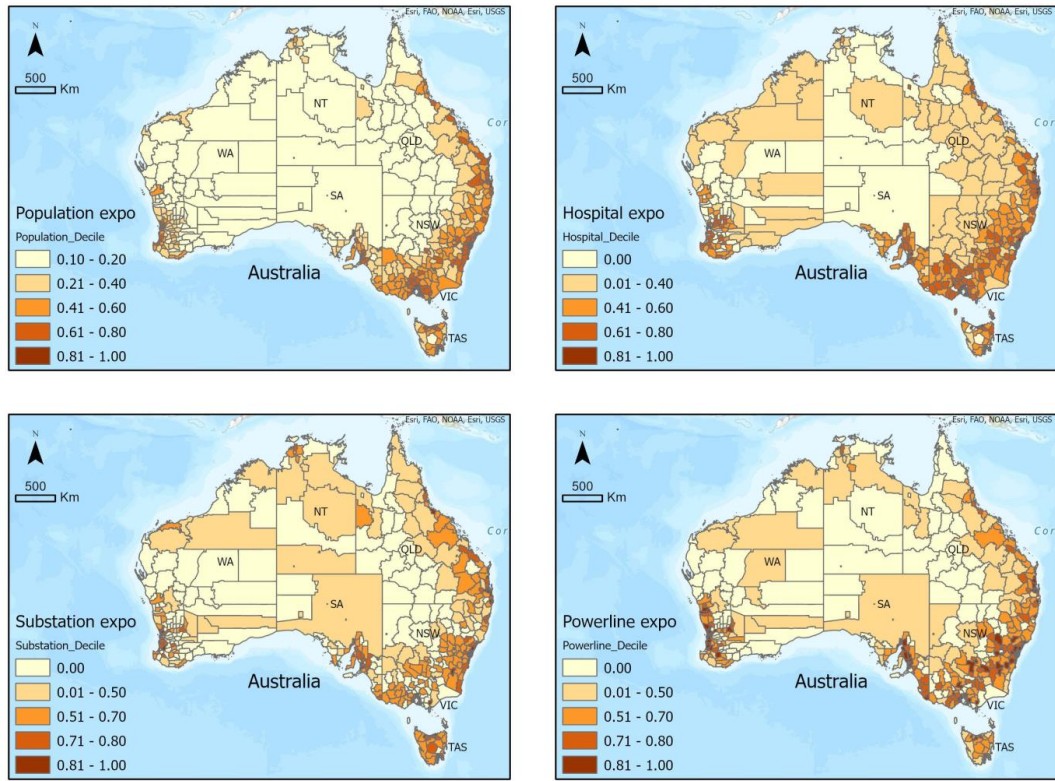

**Figure 3**. Exposure indicator maps of population, hospital, substation, and powerline density.

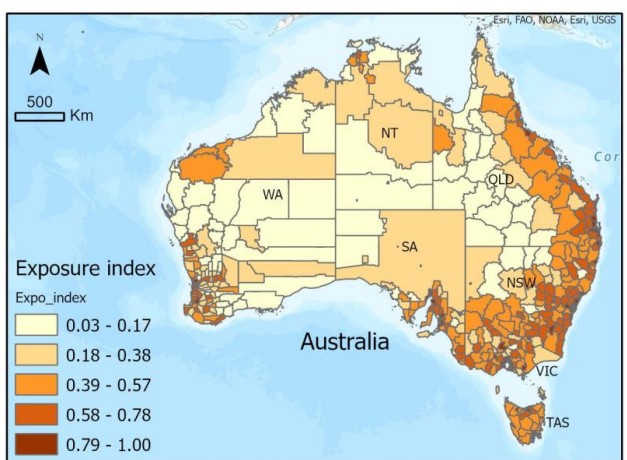


**Figure 4.** Exposure index map created by equally weighting four exposure indicators.

Exposure maps largely reflect the disproportionate percentage of Australia's population that lives on
the coast (Abuodha and Woodroffe, 2006) and near major coastal cities. As infrastructure such as
public hospitals and substations are positioned to meet demand, it is also understandable why
similar patterns are found amongst chosen indicators.

Aside from these highly populated and built-up coastal regions, relatively higher exposure index
values were identified around the Pilbara and Mount Isa regions. The mining industry's presence in
regional Australia is most obvious within the Pilbara region of north-west WA and the Mount Isa
region of north-west QLD. There are a large number of fly-in-fly-out workers for these regions and
they make a significant contribution to the economy. Although none of the chosen indicators were
mining industry-related, the population and substation densities were able to indicate significant
exposure in those areas related to the mining sector.

### 3.2. Vulnerability

The vulnerability index was created by Pareto ranking the three indicators: IRSD, vulnerable age
groups and no vehicle homes.

Figure 5 shows that IRSD vulnerability is extremely high across most of central and western Australia,
with the highest class values across almost all of Northern Territory (NT). Otherwise, vulnerability is
considerably lower in the LGAs surrounding the major cities in each state. Conversely, the vulnerable
age indicator shows the lowest values across central and western Australia. Although inner cities
also show low vulnerable age values, the highest values are found in outer suburban LGAs. For no
vehicle homes, central and north-western Australia have the highest vulnerability values, with lower
values near and surrounding major cities. The calculated vulnerability index in Figure 6 shows low to
medium vulnerability values LGAs surrounding cities, with higher vulnerability regions across NT,
northern QLD, and northern NSW to northern VIC.

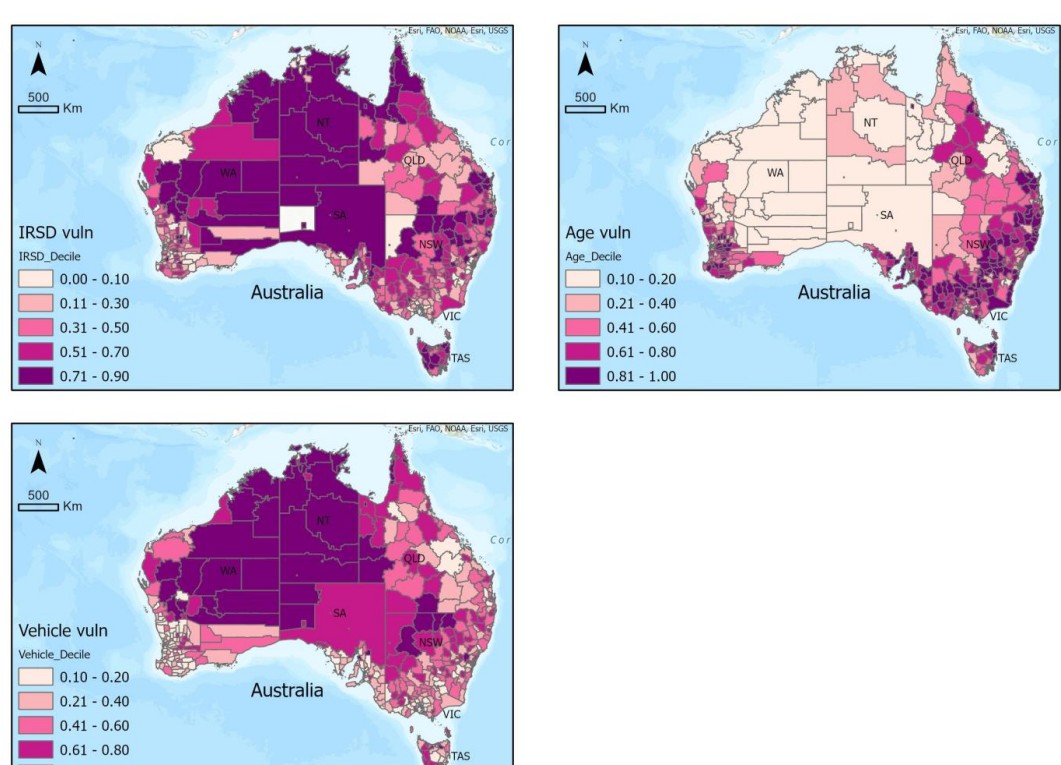

Figure 5. Vulnerability indicator maps of IRSD, vulnerable age groups and no vehicle homes.

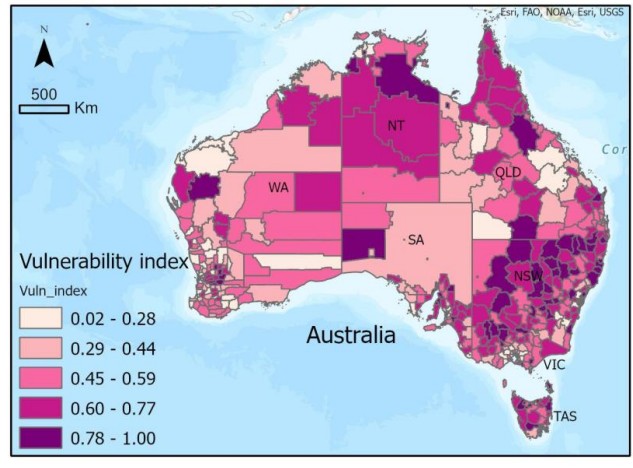


Figure 6. Vulnerability index maps calculated by Pareto ranking three vulnerability indicators.



IRSD patterns show lower vulnerability in major cities, as they are most developed and relatively affluent. High vulnerable age group values outside of and surrounding major cities can be explained by the >65 age group retiring and relocating out of urban areas (Vintila, 2001). Of the 16 IRSD input

variables, 'NOCAR', was described as the percentage of occupied private dwellings with no car. Although it is not certain whether this variable is the same as the no vehicle homes indicator used in this study from the Number of Motor Vehicles census record, some overlap is to be expected. This means regions with high no vehicle home vulnerability values are likely to have their vulnerability index overestimated. The fact that NOCAR is only one of 16 variables in the IRSD also suggests

similarities between the two indicators may be from correlation in other variables instead.

Compared to the exposure index, the transition from patterns in the indicator maps to the vulnerability index are not as clear, as Pareto ranking is used instead of equal weighting. Pareto ranking was used to address situations where a high value in one indicator would be overlooked after being equally weighted with indicators with medium to low values. Instead, it ranks LGAs on

the higher end if a single indicator's value causes it to be non-dominated much earlier. However, our analysis showed that having one indicator with the highest classification value does not guarantee a high vulnerability index value. In fact, having two indicators with the highest classification values does not guarantee a high value either as can be seen across central and north WA. This is partly because within each coloured class, there is a range of values, and only the highest values are picked

out by Pareto ranking as non-dominated. This suggests the second highest class of values in the vulnerability index (2$^{nd}$ darkest purple) are also important and possibly underestimated.

This idea of there being a lot of competition at the higher value range within indicators is highlighted by the case of the Maralinga Tjarutja LGA in western SA, which is in the highest vulnerability index class. The LGA does not have a recorded IRSD value from the ABS, meaning the region isn't

competing for a non-dominated spot on the IRSD axes. This allows the LGA to receive a very high vulnerability index score from only a very high vulnerability value in the no vehicle home indicator alone.

Overall, the vulnerability index shows higher vulnerability and thus predicts higher risk throughout NSW, northern QLD and northern NT.

3.3. TC Hazards

Hazard maps were created from datasets of chosen hazards of storm surge, flooding, wind and landslides as shown in Figure 7.

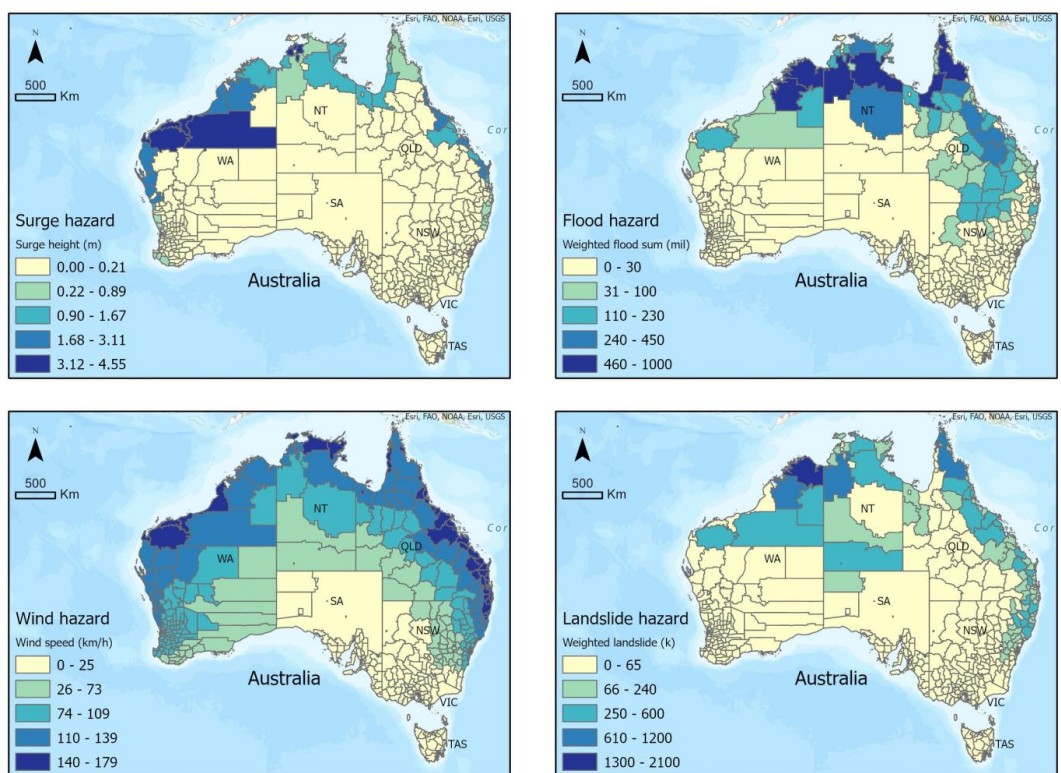

**Figure 7**. Hazard maps of storm surge, flood, wind and landslides associated with Tropical Cyclones.

Surge heights are seen to be highest in north-western WA and surrounding Darwin, having 100-year
return period surge heights greater than 3m. Surge hazard is otherwise lower around other parts of
the country's northern shoreline and has values of 0 in LGAs not bordering the coastline. Flood
hazard is shown to have the highest values across northern LGAs, with medium values over much of
QLD. Wind hazard is more consistent with TC wind speeds highest in coastal LGAs, with hazard
decreasing towards the centre of Australia and further south. Landslide hazard is highest in northern
WA along with medium values throughout NT and along the Great Dividing Range along the eastern
coast of the country.

While it would be expected that the multiple hazards associated with TCs follow the general location
TCs more commonly make landfall, there are clear differences between hazard maps in Figure 7. This
shows how the physical characteristics of each LGA can change the intensity with which different
hazards impact different regions. For example, flood and landslide hazards have the potential to
affect more inland regions while storm surge is only relevant for coastal LGAs and wind more
uniformly decreases south and inland. These results emphasise the importance of considering the
multi-hazard nature of TCs and mapping their differing extents.

The storm surge hazard map shows greater than zero values only for coastal LGAs, however, a few
LGAs may raise concern. The first is East Pilbara, the large LGA in WA with very high surge values.





Although most of the LGA is quite far inland and would not be affected by potential storm surge, the LGA does border the coastline in its northwest corner. Due to input surge datasets having the format
of point data dotted every few kilometres along Australia's coastline and chosen methods averaging intersecting surge point data to each LGA polygon, East Pilbara was mapped with very high surge hazard. For a similar reason of input hazard data only dotting the main coastland, some island LGAs were left without a surge value and thus mapped with very low hazard. For example, Tiwi Islands north of Darwin, and Mornington Island in north-western QLD. Considering their location and the
hazard values of neighbouring LGAs, these island LGAs in the country's north potentially have medium to very high hazard values rather than none at all.

An important consideration when evaluating flood and landslide hazards is that a cumulative method was used to calculate hazard values from input datasets. Rather than taking averages over each LGA as was done for surge and wind, flood and landslide input datasets were high-resolution
raster maps with many null values. Using an averaging methodology would have described an LGA's hazard in proportion to its area, meaning larger LGAs with many flood-prone regions could still have a low flood hazard value. Instead, values were summed, meaning greater than zero hazard values meant a region had some hazard-prone regions, and high hazard values meant they had more regions prone to flooding/landslides regardless of the LGA's size. While this does mean larger LGAs
have the potential to reach higher hazard values, this method represents all possible hazards, and therefore risk, rather than underestimating it due to averaging methods.

### 3.4. TC Multi-hazard Risk

Risk maps were created by multiplying each hazard with the exposure and vulnerability indices. This produced the four hazard-specific risk maps in Figure 8, from which a total TC risk map was created
by equally weighting them as seen in Figure 9.

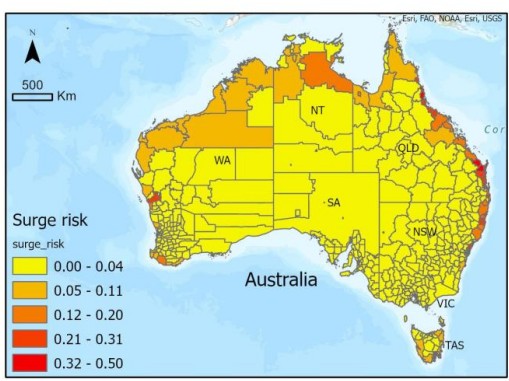

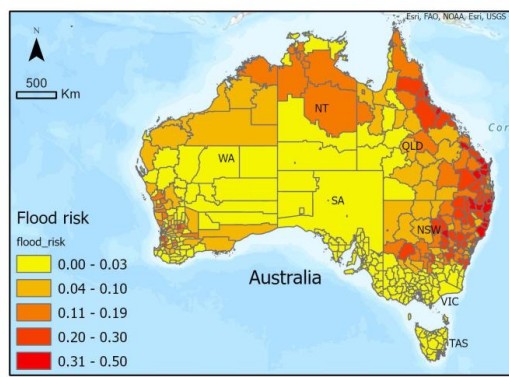

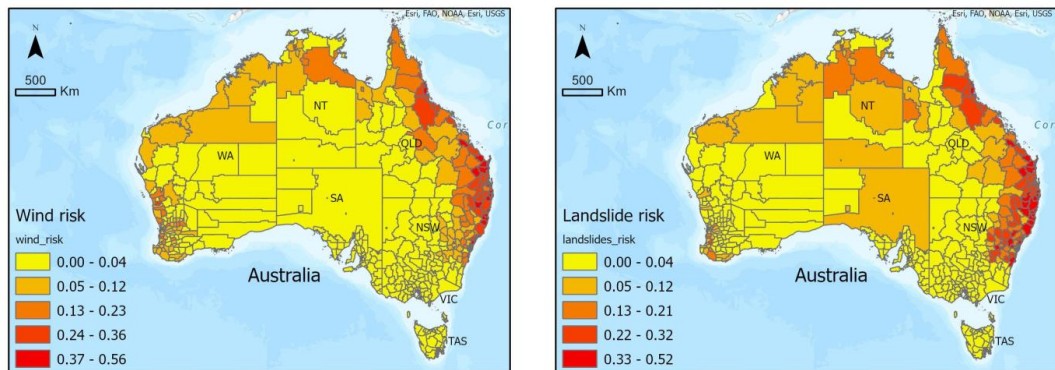

**Figure 8.** Risk maps for each hazard (surge, flood, wind, landslide).

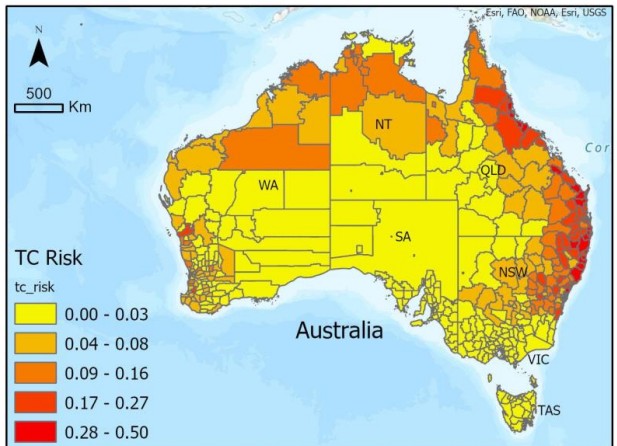

**Figure 9.** Combined multi-hazard risk calculated by equally weighting four hazard-specific risk maps.

Surge risk is considerable along the northern, western, and eastern coasts, with the highest values between Brisbane and Cairns in QLD. Flood risk can be seen to be highest across both NSW and QLD with medium values along the top of NT and WA. Risk to the wind is very uncommon at distances greater than 500 km inland and south of NSW, with the highest wind risk found along with the

eastern parts of NSW and QLD. Landslide risk also shows the highest risk in eastern NSW and QLD with medium risk across northern NT. The combined TC risk map displays some of these more prominent patterns from each hazard-specific risk map. For example, eastern NSW and QLD have the highest risk followed by medium risk across northern WA and NT. The risk to TCs is very low inland of the country surrounding SA, as well as south of NSW in VIC and Tasmania (TAS) states.

As patterns seen in risk maps can be partially explained by similar patterns found in constituent layers, it is important to compare them to hazard, exposure and vulnerability layers. While an overall TC risk map is useful for such discussions, hazard-specific risks are important to consider and


compare at a local level, for example when LGA councils are planning disaster management strategies or communicating warnings to residents for an incoming TC.

From the overall TC risk map in Figure 9, QLD and NSW have the most LGAs with very high-risk scores, particularly along the eastern coast. This result can partially be attributed to high hazard values, as well as high exposure index values with many people and infrastructure built up around those regions. Of note is that although TCs generally form over tropical waters and affect regions of Australia near the tropics, they intensify away from the equator reaching maximum intensity at

approximately 17-18°S of the equator (Kuleshov, 2020), which partially explains why risk is not highest in all northernmost LGAs. Another contributing factor to medium risk values in the country's north is due to there being relatively fewer assets exposed compared to the rest of the country, as shown by the exposure index in Figure 4. Continuing moving further south away from the tropics, TCs are weakening as sea surface temperatures get colder in extra-tropical regions. Hence,

substantial reduction of risk is observed in VIC and TAS. Similarly, TCs weaken over land which is why risk is also very low for central Australian LGAs. The lower risk in these states is supported by historical records of TC tracks from 1970-present (Kuleshov, 2020).

### 3.5. Limitations of Risk Assessment

One of the limitations of this TC risk assessment of Australian LGAs is that indicators were selected

partially because of availability, and hence may not represent all aspects of hazard, exposure, or vulnerability. For example, within the vulnerability index, indicators that informed a region's preparedness to natural disaster events were not available. While some LGA councils may have informative documents or evacuation plans, it is difficult to determine how well understood they are by residents, and the data is not standardised in the format that can be compared against LGAs

across the country. Additionally, in some cases lower resolution global hazard datasets were used because they were available, while higher resolution, Australia-specific datasets are yet to been created or were inaccessible.

Being a risk assessment, subjective indicator choices were made which can shift how results should be interpreted (Aguirre-Ayerbe et al., 2018; Brooks, 2003). For example, chosen exposure indicators

identified regions where many lives were exposed alongside physical lifeline infrastructure that contributes to health and utilities (hospitals, electricity). These indicators however do not necessarily represent potential financial losses if businesses and industries were not able to function due to TC impact. As a result, discussion of risk map implications would need to stay human-centric. While just adding more indicators could be identified as a possible solution, the nature of risk and index

calculations mean that adding more indicators reduces the importance of each, resulting in a potentially less informative final risk map.

Another limitation is that while each indicator map had patterns identified, the discussion was based on an incomplete understanding of Australian LGAs. Ideally, formal validation of each indicator with local knowledge from people who reside in or manage each LGA would ensure that each

contributing input to end risk maps were accurately represented. Engagement with indigenous people would also be an essential aspect of validation so that cultural assets and indigenous knowledge are included in the maps.


## 4. Conclusions

The developed novel methodology for multi-hazard TC risk assessment for Australia and created maps showed the differences in hazard extent and differing characteristics of each region that made an LGA at risk to TCs. Generally, the highest level for all TC-related hazards was found along the eastern, northern, and western coasts, with all hazards being weakest far inland and in the southern parts of the country. Selected exposure indicators represented human lives as the most important

asset at risk, which was found to be highest around major coastal cities in each state, while vulnerability showed more varied spatial distribution. Final TC risk maps suggested most at-risk states were QLD and NSW for all hazards, particularly in the states' eastern regions followed by medium risk across Northern Territory and north-west of Western Australia. As with all risk assessments, the selected indicators should be considered before using resultant maps to inform

decisions, and future work includes all-important validation studies.

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

**Author contribution**

Conceptualization, C.D. and Y.K.; methodology, C.D. and Y.K.; software, C.D.; validation, C.D.; formal analysis, C.D.; investigation, C.D.; resources, C.D. and Y.K..; data curation, C.D. and Y.K.; writing—original draft preparation, C.D.; writing—review and editing, C.D. and Y.K.; supervision, Y.K.; project administration, Y.K.

**Competing interests**

Authors declare not competing interest.

**Appendix**

Data table for LGA risk analysis. Links are provided for the data sources as well as the year that the dataset was last updated.

| Indicator | Dataset used | Source | Year |
|---|---|---|---|
| **Hazard** | | | |
| Surge hazard | Point feature layer of Storm surge run-up height, 100yr return period | GAR Atlas | 2015 |
| Flood hazard | Raster Flood depth inundation, 100yr return period | GAR Atlas | 2015 |
| Wind hazard | Raster Cyclone wind, 100yr return period | Geosciences Australia | 2018 |
| Landslide hazard | Raster Global landslides hazard | ARUP | 2020 |
| **LGA Exposure** | | | |




| Population density | Recorded total number of people living in each LGA. | ABS Census data | 2016 |
|---|---|---|---|
| Public hospital | Point feature layer of public hospitals around Australia | ArcGIS Online Dataset | 2019 |
| Substations | Point feature layer of power substations around Australia | Geosciences Australia | 2016 |
| Powerlines | Line feature layer of powerlines around Australia | Geosciences Australia | 2016 |
| **LGA Vulnerability** | | | |
| IRSD | Summary statistic for socioeconomic status, | ABS Census data | 2016 |
| No vehicle homes | Percentage of households within each LGA that owns zero vehicles. | ABS Census data | 2016 |
| Vulnerable age groups | Percentage of LGA population that is under 15 or over 65 | ABS Census data | 2016 |
| **Shape layers** | | | |
| LGA polygon layer | Shapefile containing the size of each LGA as of 2016 | ABS | 2016 |
