# Peer review of "Multi-hazard Tropical Cyclone Risk Assessment for Australia"

_Natural Hazards and Earth System Sciences, 2022_

## Author Comment (AC1)

**Document addressing reviewer comments**

Dear editor, I have read accurately the manuscript and I found some criticalities that, in my opinion, should be solved before considering it for publication. In the following you can see my comments for the Authors.

5   Dear Authors, I have read accurately your manuscript and I found some criticalities that, in my opinion, should be solved.

Dear Reviewer, thank you for your valuable comments which helped us to improve quality of the manuscript. All your comments have been addressed in a revised version of the manuscript. We hope you will find this revision satisfactory.

10   First of all, at the beginning of the manuscript could be useful a table with the acronyms used in the text.

Addressed. An acronym table has been added below the abstract.

Section 1

Line 29: please specify the scale of intensity of TCs.

15   Addressed. The wind speed threshold for the 4-5 categories mentioned has been provided along with the scale used (Saffir-Simspon scale).

Lines 42-47: I maintain that could be useful a map showing the LGAs and Australian States (WA, NT, SA, QLD, NSW, VIC, and TAS) and some statistics on LGAs (e.g. number of LGAs, minimum, maximum, mean, and median extension). I suggest adding a study area section
20   describing the physical and economic characteristic of Australia and all the toponyms cited in Section 3. Moreover, you do not explain at all the TC risk in Australia. It is important to add a part (may be in the study area section) describing the TC risk in Australia.

Addressed. A study area section has been added, with a map of Australian states and territories, major cities as well as a general characteristics table of each state.

25   Lines 53-68: It is not clear if in your study the analysed risks are caused by TCs. It is a cascade approach or not? I do not understand why and how it is possible to obtain TC risk combining surge, flood, wind, and landslide risks.

Addressed. This is clarified in the revised manuscript, specifying it is referring to TC-induced surge/flood/wind/landslide. All four TC-induced hazards are used in the calculation of their
30   hazard specific risk (e.g. TC wind risk, TC flood risk, etc.).

Section 2

In my opinion in this section must be added a short part describing the difference among variables, indicators, and indexes. May help citing the "Pyramid of Information" of Hammond et al. (1995) or explaining better the IRDS that you cite in your paper.

35   Addressed. The three tiers/stages of create a risk index are detailed in new figure 3, with additional paragraph.

Moreover, I maintain that the table in Appendix may be reported in Section 2 and I suggest adding in the table the data format and resolution. In addition, I suggest explaining better these data in Section 2.1.

Addressed. The appendix data table has been moved into section 2 and has a new column with data format and resolution.

Lines 70-75: If I understand correctly, you started from hazard, exposure, and vulnerability indicators that were combined to obtain hazard, exposure, and vulnerability indexes by using equal weighting for exposure and Pareto front-ranking for vulnerability (and for risk?). Consequently, I suppose, you obtained surge, flood, wind, and landslide risks. And, finally, combing these risks you obtained TC risk. It is correct? Please explain it better. In my opinion the sentence "This data was then joined to LGA map shapefiles in ArcGIS Pro" may be changed in "This data was then combined to LGA map shapefiles in ArcGIS Pro", because "join" is a particular GIS command.

Correct. This is clarified with the new diagram explaining the process. GIS jargon terms have been replaced with more general terms.

I maintain that Figure 1 do not explain in a correct way the risk mapping process. I suggest separating the part of the figure concerning the four term of equation 1 (risk, hazard, exposure, and vulnerability), as well as the different considered risks (surge, flood, wind, landslide, and TC). Moreover, the figure does not explain clearly the process of transforming the indicators to indexes. I suggest modifying the figure and explaining it analytically. Additionally, I suggest explaining briefly the Pareto ranking and the Figure 2 that, in my opinion, it is not clear. Consequently, I suggest re-writing the Section 2.3.

Addressed. Old Figure 1 has been removed and replaced with Figure 3 which separates each process. More accompanying paragraphs are added to walk readers through the process.

Section 3

The acronyms of the Australian states are not always evident in the maps. Please modify the maps accordingly.

Addressed. Section 2 (study area) has been added with Figure 1 which labels the boundaries of each state and territory.

You cited Pibara region, Mount Isa LGA, major coastal cities, major cities, inner cities, mining industries, urban areas, Maralinga Tjarutja LGA, Darwin, Great Diving Range, Tiwi and Mornigton Islnds, Bisbane, Cairns. Where are located these areas?

Addressed. Maps have now been annotated on most relevant figure where these locations are referenced (Pilbara, Maralinga Tjarutja, Townsville, Tiwi Islands, Mornington Island). Major cities have been highlighted in Figure 1. Some previous location names have been swapped out for descriptions of their location instead.

Overall, in my opinion the manuscript needs to be improved before considering it for publication.

Addressed. Major revision was undertaken to improve results and presentation.

**Multi-hazard Tropical Cyclone Risk Assessment for Australia**

Cameron Do and Yuriy Kuleshov

Natural Hazards and Earth System Sciences https://www.natural-hazards-and-earth-system-sciences.net/

**Abstract**

Tropical cyclones (TCs) have long posed a significant threat to Australia's population, infrastructure, and environment. This threat may grow under climate change as projections indicate continuing sea level rise and increases in rainfall during TC events. Previous TC risk  assessment efforts have focused on the risk from wind, whereas a holistic approach requires multi-hazard risk assessments that also consider impacts of other TC-related hazards. This study assessed and mapped TC risk nationwide, focusing on the impacts on population and infrastructure from the TC-related hazards of wind, storm surge, flooding and landslides. Risk maps were created at the Local Government Area (LGA) level for all of Australia, using collated data on multiple hazards, exposure and vulnerability. The results  demonstrated that the risk posed by all hazards was highest for coastal LGAs of eastern Queensland and New South Wales followed by medium risk across Northern Territory and north-west of Western Australia. Further enhancement  and validation of risk maps developed in this study will provide decision-makers with the information needed to reduce TC risk, save lives and prevent damage to infrastructure.

**Acronyms**

| | |
|---|---|
| ABS | Australian Bureau of Statistics |
| LGA | Local Government Area |
| NSW | New South Wales |
| NT | Northern Territory |
| QLD | Queensland |
| SEIFA | Socio-Economic Indexes For Areas |
| SA | South Australia |
| TAS | Tasmania |
| TC | Tropical Cyclone |
| VIC | Victoria |
| WA | Western Australia |

**Comment [CD1]:** Reviewer: acronym table

**1. Introduction**

Tropical Cyclones (TCs), also known as hurricanes or typhoons, are powerful and highly destructive meteorological hazards. Since 1970, almost 2,000 natural disasters have been attributed to TCs, which has led to over 700,000 deaths worldwide (World Meteorological Organisation, 2021). Costing about U.S.$26 billion annually in global damages (Mendelsohn et al., 2012), their impact is expected to multiply to U.S.$60 billion annually by 2100 (Bakkensen and Mendelsohn, 2019).

105 The proportion of intense TCs (Saffir–Simpson scale categories 4-5 with  1-minute maximum sustained winds >209km/h) and peak wind speeds of the most intense TCs are projected to increase at the global scale with increasing impact of global warming (high confidence) (IPCC, 2022)  1). The potential of more destructive TC events will require updating and enhancement of existing risk reduction strategy. The Sendai Framework for Disaster Risk Reduction provides a structure for reducing disaster damages and increasing resilience to hazards

110 including TCs (Bennett, 2020). One mechanism they encourage in Goals 18 and 24 is the distribution of multi-hazard risk information such as risk assessments.

Risk assessments combine hazard information with human activity, infrastructure and natural resources to determine the possible impacts of hazardous events (Belluck et al., 2006; National Research Council, 1991) and make informed choices for risk management in the most exposed and

115 vulnerable regions (Aguirre-Ayerbe et al., 2018). Disaster risk is defined as the probability of harmful consequences, or significant losses, resulting from interactions between a hazard, and the local exposure and vulnerability to that hazard (Crichton, 1999; Downing, 2001).

As Local Government Areas (LGAs) are  one of the smallest government decision-making bod with available census data, information is sought to be provided on that scale. Risk assessments are

120 a foundation for early warning systems to raise alerts of potential impacts, and to provide evidence for the prioritisation of funds and resources to areas in advance of any hazardous events. While the climate continues to change alongside evolving human activity, risk assessments must likewise be regularly updated to stay accurate and useful as a tool for disaster risk reduction (Peduzzi et al., 2012).

125 For TCs, the four main hazards are the destructive winds, associated storm surge, flooding from associated heavy rainfall, and landslides on steep terrain as soils saturate (Murray et al., 2020). TCs and other natural hazards are becoming increasingly recognised as multi-hazardous in nature (Scawthorn et al., 2006) . These hazards impact regions differently and their effects can compound to cause even greater damage (Gori et al., 2020).

130 While TCs can cause damage through different hazards, such as gale-force winds, storm surge or flooding, the communication of TC intensity and categorisation places emphasis on wind speed (Lavender and Mcbride, 2020). This is partially due to the availability of wind measuring technology and the relative ease to quantify wind. Publicly available warnings and forecasts are focusing on wind speeds, ultimately portraying the message that winds are the hazard to be most wary of. The

135 literature however suggests the TC-induced impacts of storm surge and flooding contribute to the most human lives lost and infrastructure damage (Mendelsohn et al., 2012; Zhang et al., 2008). Although some studies have included multi-hazard aspects of TCs (Burston et al., 2017) , presenting different hazard models for TC-induced storm surge, wind and flooding, these studies do not complete the story of combining hazard with exposure and vulnerability to map risk.

140 Similarly, within the literature, there are many examples of standalone exposure or vulnerability index assessments for TCs (Marín-Monroy et al., 2020; Bathi and Das, 2016; Amadio et al., 2019). This gap indicates compelling scope to develop a multi-hazard TC risk assessment that can differentiate the extent and severity of TC- induced hazards.

This study will address this gap and strengthen TC risk information for the Australian region. Multi-

145 hazard risk is assessed and visualised through  maps which show LGA categorisation,

alongside hazard, exposure, and vulnerability layers. As a risk assessment's usefulness relies on how they are tailored for a specific users audience or applications, the method proposed in this study serves as a proof of concept that can be altered in future iterations for tailored use.

**2. Study area**

Figure 1. Map of study area, state, and territory boundaries as well as Local Government Area (LGA) divisions and major cities. States and territories: Western Australia (WA), Northern Territory (NT), Queensland (QLD), South Australia (SA), New South Wales (NSW), Australian Capital Territory (ACT), Victoria (VIC) and Tasmania (TAS) are labelled.

Table 1. Comparison table of each Australian states general characteristics including total area, real GSP and population (Australian Bureau of Statistics, 2020-2021)

| STATE | TOTAL AREA | Number of LGAs | Avg. area per LGA | GSP ($million) | Population | GSP per capita ($) |
|---|---|---|---|---|---|---|
| Australia Capital Territory (ACT) | 2358 | 1 | 2358.172 | 433740 | 431483 | 100523 |
| New South Wales (NSW) | 800811 | 130 | 6160.083 | 6336350 | 8172561 | 77532 |
| Northern Territory (NT) | 1348094 | 18 | 74894.13 | 261810 | 246565 | 106183 |
| Queensland (QLD) | 1730172 | 78 | 22181.69 | 3689770 | 5194884 | 71027 |
| South Australia (SA) | 984275 | 71 | 13863.03 | 1149210 | 1770794 | 64898 |
| Tasmania (TAS) | 68018 | 29 | 2345.443 | 340830 | 541499 | 62942 |
| Victoria (VIC) | 227496 | 80 | 2843.695 | 4682640 | 6661697 | 70292 |
| Western Australia (WA) | 2526646 | 137 | 18442.67 | 3206530 | 2670231 | 120084 |

Australia is a country with a long coastline and with much of its northern states commonly impacted by tropical cyclonesTCs. An average of 12 TCs form in the Australian region annually (however,

interannual variability is high ranging from 19 TCs in 1983/84 to 3 TCs in 2015/16, for records examined from 1970/71 to 2019/20 TC seasons (Kuleshov et al., 2020)), with 5 making landfall on average (Mortlock et al., 2018). In the last few decades, several severe TC events have destroyed infrastructure and caused billions of dollars in losses, including TC Larry (2006), TC Yasi (2011) and TC Debbie (2017).

Figure 1 shows the boundaries of each state and territory as well as the outline of Local Government Area (LGA) divisions within. Table 1 summarises key traits of each state such as their total area, real Gross State Product (GSP) and population. From the Ttable 1 it can be seen that NSW and VIC are the states with the highest GSP (monetary measure of state output), as well as highest populations. For TC-related impacts however, we are most concerned with interested in the northern states that are expected to more commonly be impacted by TC events. QLD and WA therefore stand out as the next most important states with next highest GSP and populations. Important to note however is the size of QLD and WA states and much higher average area per LGA, meaning GSP contribution and populations are likely to be much more spread out.

**2.3. Data and Methodology**

To calculate the multi-hazard risk of TCs to Australia, hazard, exposureexposure, and vulnerability datasets were chosen and sourced. This data was then joined combined to LGA map shapefiles in ArcGIS Pro. To calculate exposure and vulnerability indexes from multiple indicators, equal weighting was used for exposure, while Pareto front-ranking was used for vulnerability. Combined with hazard values for each LGA, exposure and vulnerability indexes were used to calculate risk using equation 1:

$$\text{Risk} = \text{Hazard} \times \text{Exposure} \times \text{Vulnerability} \qquad (1)$$

**1.1. 3.1 Selection of indicators**

Table 2. Data table for LGA risk analysis. Links are provided for the data sources as well as the year that the dataset was last updated.

| Indicator | Dataset used | Source | Year | Data format and resolution |
|---|---|---|---|---|
| **Hazard** | | | | |
| Surge hazard | Global tropical cyclone storm surge run-up height, 100yr return period | GAR Atlas (2015) | 2015 | Point data (every 1km along coastline) |
| Flood hazard | Australian flood depth inundation, 100yr return period | GAR Atlas (Rudari et al., 2015) | 2015 | Raster data (1km) |

| | | | | |
|---|---|---|---|---|
| Wind hazard | Australian tropical cyclone wind, 100yr return period | Geoscience Australia (Arthur, 2018) | 2018 | Raster data (2km) |
| Landslide hazard | Global landslides hazard | ThinkHazard! (Arup, 2020) | 2020 | Raster data (1km) |
| **LGA Exposure** | | | | |
| Population | Recorded total number of people living in each LGA. | ABS Census data | 2016 | Tabular data (LGA resolution) |
| Public hospital | Point feature layer of public hospitals around Australia | ArcGIS Online Dataset | 2019 | Point data |
| Substations | Point feature layer of power substations around Australia | Geoscience Australia | 2016 | Point data |
| Powerlines | Line feature layer of powerlines around Australia | Geoscience Australia | 2016 | Line data |
| **LGA Vulnerability** | | | | |
| IRSD | Summary statistic for socioeconomic status, | ABS Census data | 2016 | Tabular data (LGA resolution) |
| No vehicle homes | Percentage of households within each LGA that owns zero vehicles. | ABS Census data | 2016 | Tabular data (LGA resolution) |
| Vulnerable age groups | Percentage of LGA population that is under 15 or over 65 | ABS Census data | 2016 | Tabular data (LGA resolution) |
| **Shape layers** | | | | |
| LGA polygon layer | Shapefile containing the size of each LGA as of 2016 | ABS Australian Statistical Geography Standard | 2016 | Polygon data (LGA resolution) |

185

**Hazard**

The main identified hazards of TCs include storm surge, winds, landslides, and floods. The 100-year return period was chosen to represent the current long-term probability  of these hazards occurring.  Of note is that these probabilities may change in the future with studies predicting increased storm surge levels (high confidence) and increased TC-related precipitation (medium-high confidence) due to climate change (Cha et al., 2020).

Storm surge data was acquired from GAR Atlas' risk and data platform, which mapped TC storm surge height as point data roughly along the Australian coastline every 1km. TC wind data was sourced from Arthur (2018) and came as high-resolution raster data over Australia and its northern

195 waters. Flood data was sourced from Rudari et al. (2015) as high-resolution raster data representing riverine flooding only. Thus, non-null values tended to only appear near riverine systems and catchments. Similarly, landslide data from Arup (2020) was in the raster format with mostly null values apart from specific locations with significant landslide hazard.

Storm {Arthur, 2018 #181@@author-year}and wind datasets were specifically designed for TCs

200 (Cardona et al., 2014; Arthur, 2021)surge and wind, and spatial mean values were calculated over each LGA. For flood and landslide hazards the original datasets did not consider solely TC induced floods/landslides. Thus, the flood and landslide hazards were weighted towards TC prone regions by multiplying values by the TC wind raster dataset. Weighted flood and landslide values were then summed over LGAs as there were many null values. Greater than zero values exist only around water

205 catchments and rivers for floods, and around mountain regions for landslides. Thus, LGAs with higher flood and landslide values have more of these prone environments in total rather than a higher areal proportion.

**Exposure**

Exposure indicators of population, hospitals, substations, and power lines were chosen to represent

210 physical assets of human life, as well as systems and infrastructure that are important in the case of emergency disaster events (hospitals, power). Failure to maintain the function of lifeline infrastructures such as hospitals and power can lead to exacerbated negative impacts (Ju et al., 2019). These chosen indicators aim to spatially describe which LGA regions have more exposed assets relative to the rest of the country. Electrical substations provide power as critical

215 infrastructure and are strategically placed to meet demand. Similar reasoning influenced the choice of public hospitals and powerlines.

While population density data of each LGA was found in tabular form from the Australian Bureau of Statistics (ABS), the remaining exposure indicators' raw format was as point or line shapefiles displayable in ArcGIS Pro. Thus geoprocessing tools such as spatial join were used to count the

220 number of public hospitals in each LGA. Using absolute measurements can be inappropriate when considering regions of different sizes (Rygel et al., 2006), thus these counts were then divided by LGA area to give a density value similar to that of population density.

**Vulnerability**

Vulnerability indicators were chosen to represent regions most susceptible to high impact from a TC

225 event occurring in the vicinity. Measures of socioeconomic status are commonly used to describe vulnerability to natural hazard events (Mitsova et al., 2018; Lianxiao and Morimoto, 2019) and the Index of Relative Socioeconomic Disadvantage (IRSD) has been used in earlier studies previous literature for the Australian region (Rolfe et al., 2020). It summarises variables about the social and economic conditions of households. The more disadvantaged a region is socioeconomically, the

230 more likely it will be more impacted by TCs, due to factors such as lower income, having families with only one parent or having a higher percentage of people that have English as a second language. The 'no vehicle homes' indicator was derived by calculating the percentage of homes with

no vehicles, and the 'vulnerable age groups' indicator was constructed by calculating the percentage of an LGA's population made up of the <15 and >65 age group combined. The 'no vehicle homes' indicator is considered as  relevant to TCs as it provides information on LGAs that are more susceptible to loss of human life in evacuation situations. Although this risk assessment highly values human life and safety, historically within Australia, TCs have caused very few fatalities in recent decades, and an indicator describing the vulnerability of infrastructure would be preferred. An alternative to 'no vehicle homes' vulnerability indicator could be the proportion of houses that are not constructed to modern wind loading standards. While this potentially useful indicator was not included in this study due to limited data availability, this could be a topic for future work.

Vulnerability indicators are ideally directly linked to their relevant exposed counterparts; however, these human and society-centred vulnerability indicators were chosen to generally relate to selected  exposure indicators which can be estimated as populations and the built environment they are surrounded by. Direct infrastructural vulnerability indicators were of interest, such as building code standards to give information on their susceptibility to wind damages, however due to limited data and the multi-hazard approach of this study, a more general approach was taken.

1.2. 3.2 Mapping Process

First, acquired indicator data was transformed and converted into the LGA resolution. As raw data came in tabular (a), point (b) and raster (c) formats, different methods for each were used to summarise information when converted to LGA polygons (d) as depicted in Figure 2. Tabular data from the ABS came at an LGA resolution, so data only needed to be linked to an LGA polygon shapefile in ArcGIS Pro. For storm surge point data which spanned across the coastline every 1km, the average 100-yr surge height value was taken, whereas for exposure point data such as hospitals and substations, the count or number of points in each LGA was taken. For the wind raster data which had no null values and gradually changed in value inland, the mean windspeed value was taken per LGA, while with flood and landslide data the sum of non-null values was taken per LGA.

> **Comment [CD2]:** Reviewer: move appendix up here - add columns on data format and resolution
>

[Figure]

**Figure 2**. Diagram representing data formats of acquired raw data (tabular (a), point (b), raster (c)) being transformed into a comparable LGA polygon format (d). (Example data is used here, and (d) is not representative of any results)

Once in a comparable data format, indicator data values were normalised into a 0 to 1 range with decile normalisation against the whole country using Python scripts. Use of different normalisation methods were tested, such as linear normalisation and natural breaks, however decile normalisation was found to best remove the skewing effects of outliers, and is a method commonly used in several ABS indices.

Figure 3 depicts the different tiers or stages of the risk assessment, starting at tier (3) with the indicators. These are the variables that differ in value spatially across Australian LGAs, that were chosen to be representative of TC hazard, exposure, or vulnerability. Three to four indicators were chosen to give a more robust index without diluting the sensitivity of each indicator. From tier (3) to tier (2), or from indicators to indices, different methods were used depending on the index. For exposure, equal weighting was used, while Pareto ranking was used for vulnerability, which is explained in the  sub-section (3.3). With only one indicator or dataset for each hazard of TCs, each hazard was passed through separately, meaning when hazard, exposure, and vulnerability indices were multiplied using equation 1 to calculate risk (tier (2) to tier (1)), four hazard specific risk layers were produced - TC-induced wind risk, flood risk, surge risk and landslide risk. A quantile classification with five classes (ten for hazard) was used for map symbology.

265

270

275

280

**Comment [CD3]:** Delete

[Figure]

Figure 3. Risk index flowchart from tier (3) indicators to hazard, exposure and vulnerability indices in tier (2), then finally the final risk index (1). Four hazard-specific risk layers are produced, based on the chosen TC-induced hazard indicator chosen.

Transforming raw indicator and hazard data into a workable file format
(e.g., spatially representing data on a map as shapefiles, preparing raster hazard data)

↓

Summarising relevant values for each LGA (e.g., averaging wind hazard cell values,
counting number of hospitals)

↓

Processing summarised values (e.g., converting the number of hospitals to hospitals per sq. km)

↓

Normalising values into a 0 to 1 range by ranking each LGA by decile against the rest of the country

↓

Creating the exposure index by equal weighting (averaging) decile values
of the four exposure indicators

[Figure]

Creating the vulnerability index by using Pareto Front ranking on the three vulnerability indicators

Creating the risk index for each hazard by multiplying decile hazard values, the exposure index, and the vulnerability index for each LGA

Creating map layers for each hazard, each indicator and each index, colour coding

and using five classes of natural breaks as the symbology classification

Creating a final TC risk layer by equally weighting the risk of each hazard

**Figure 1**. TC risk mapping process.

**2.3. 3.3 Indices Calculation**

[revised manuscript text omitted]

Aside from these highly populated and built-up coastal regions near major cities, relatively higher exposure index values were identified around the Pilbara and Mount Isa regionsTownsville regions. The mining industry's presence in regional Australia is most obvious within the Pilbara region of north-west WA, with and the Mount Isa region of north-west QLD. There are a large number ofmany fly-in-fly-out workers for these regions, and they which make a significant contribution to the economy. Although none of the chosen indicators were mining industry-related, the population

405 and infrastructure indicators were able to detect significant exposure in those areas related to the mining sector. High exposure of Townsville can be explained by a high population making it the largest settlement in North QLD, along with moderate infrastructure indicator values. The city is a popular tourist destination being adjacent to the Great Barrier Reef and national parks, while also hosting large metal refineries.

410 From the  exposure maps, we can see that assets possibly lost in a TC event are highest along the country's eastern coastline as well as surrounding major cities. Thus we would expect the potential for highest risk in these regions if vulnerability and hazard are also high.

 4.2 Vulnerability

The vulnerability index was created by Pareto ranking the three indicators: IRSD, vulnerable age
415 groups and no vehicle homes.

Figure 7 shows that IRSD vulnerability is extremely high across most of central and western Australia, with the highest class values across almost all of NT. Otherwise, vulnerability is considerably lower in the LGAs surrounding the major cities in each state. Conversely, the vulnerable age indicator shows the lowest values across central and western Australia. Although inner cities
420 also show low vulnerable age values, the highest values are found in outer suburban LGAs. For no vehicle homes, central and north-western Australia have the highest vulnerability values, with lower values near and surrounding major cities. The calculated vulnerability index in Figure 8 shows low to medium vulnerability values in LGAs surrounding cities, with higher vulnerability regions across NT, northern QLD, and northern NSW to northern VIC.

425

[Figure]

[Figure]

[Figure]

[Figure]

[Figure]

[Figure]

430 **Figure 75**. Vulnerability indicator maps of IRSD, vulnerable age groups and no vehicle homes.

[Figure]

[Figure]

**Figure** 8. Vulnerability index maps calculated by Pareto ranking three vulnerability indicators.

IRSD patterns show lower vulnerability in major cities, as they are most developed and relatively
435    affluent. High vulnerable age group values outside of and surrounding major cities can be explained
by the >65 age group retiring and relocating out of urban areas (Vintila, 2001). Of the 16 IRSD input
variables, 'NOCAR', was described as the percentage of occupied private dwellings with no car.
Although it is not certain whether this variable is the same as the no vehicle homes indicator used in
this study from the Number of Motor Vehicles census record, some overlap is to be expected. This
440    means regions with high no vehicle home vulnerability values are likely to have their vulnerability
index overestimated. The fact that NOCAR is only one of 16 variables in the IRSD also suggests
similarities between the two indicators may be from correlation in other variables instead.

Compared to the exposure index, the transition from patterns in the indicator maps to the vulnerability index are not as clear, as Pareto ranking is used instead of equal weighting. Pareto ranking was used to address situations where a high value in one indicator would be overlooked after being equally weighted with indicators with medium to low values. Instead, it ranks LGAs on the higher end if a single indicator's value causes it to be non-dominated much earlier. However, our analysis showed that having one indicator with the highest classification value does not guarantee a high vulnerability index value. In fact, having two indicators with the highest classification values does not guarantee a high value either as can be seen across central and north WA. This is partly because within each coloured class, there is a range of values, and only the highest values are picked out by Pareto ranking as non-dominated. This suggests the second highest class of values in the vulnerability index ($2^{nd}$ darkest purple) are also important and possibly underestimated.

This idea of there being a lot of competition at the higher value range within indicators is highlighted by the case of the Maralinga Tjarutja LGA in western SA, which is in the highest vulnerability index class. The LGA does not have a recorded IRSD value from the ABS, meaning the region isn't competing for a non-dominated spot on the IRSD axes. This allows the LGA to receive a very high vulnerability index score from only a very high vulnerability value in the no vehicle home indicator alone.

Overall, the vulnerability index shows higher vulnerability and thus predicts higher risk throughout NSW, northern QLD and northern NT.

**2.3. 4.3 TC Hazards**

TC hHazard maps were created from datasets of chosen hazards of storm surge, flooding, wind, and landslides as shown in Figure 97. Ten quantile classes (decile) were used to present these TC hazard maps to represent values and display any trends more precisely.

[Figure]

[Figure]

[Figure]

[Figure]

[Figure]

[Figure]

**Figure 9~7~.** Hazard maps of storm surge, flood, wind and landslides associated with Tropical Cyclones.

Surge heights are seen to be highest along a long portion of~in~ north-western WA,~and~ surrounding Darwin, and at a few locations along QLD's eastern coastline, having 100-year return period surge heights greater than 2~3~m. Surge hazard is otherwise lower around other parts of the country's northern shoreline and has values of 0 in LGAs not bordering the coastline. Flood hazard is shown to have the highest values across northern ~LGAs~WA, NT and most of QLD, with ~medium~significant values ~over much of QLD~reaching into NSW. Wind hazard is more consistent, with TC wind speeds highest in coastal LGAs, with hazard decreasing towards the centre of Australia and further south. Landslide hazard is highest in northern WA and NT along with ~medium~ high values throughout ~throughout NT and along the Great Dividing Range along the eastern coast of the country~eastern QLD and north eastern NSW.

While it would be expected that the multiple hazards associated with TCs follow the general location TCs more commonly make landfall, there are clear differences between hazard maps in Figure 9~7~. This shows how the physical characteristics of each LGA can change the intensity with which different TC hazards impact different regions. For example, flood and landslide hazards have the potential to affect more inland regions while storm surge is only relevant for coastal LGAs, and wind more uniformly decreases south and inland. These results emphasise the importance of considering the multi-hazard nature of TCs and mapping their differing extents.

The storm surge hazard map shows greater than zero values only for coastal LGAs, however, a few LGAs may raise concern. The first is East Pilbara, the large LGA in WA with very high surge values. Although most of the LGA is quite far inland and would not be affected by potential storm surge, the LGA does border the coastline in its northwest corner. Due to input surge datasets having the format of point data dotted every few kilometres along Australia's coastline and chosen methods averaging intersecting surge point data to each LGA polygon, East Pilbara was mapped with very high surge hazard. For a similar reason of input hazard data only dotting the ~main coastland~mainland, some island LGAs were left without a surge value and thus mapped with ~very low~lowest hazard. For example, Tiwi Islands north of Darwin, and Mornington Island in north-western QLD. Considering their location and the hazard values of neighbouring LGAs, these island LGAs in the country's north potentially have medium to very high hazard values rather than none at all. These cases pose the question of the chosen LGA resolution in this study, with higher resolutions being preferred

especially for hazard indicators. The method in this study however applies the same rules and calculations for all LGAs, which allows for a low resource and quick rendition of relative hazard.

Wind hazard trends are to be expected and are consistent with TC genesis and development  where TCs start to  lose intensity after landfall, having its energy source of warm ocean water cut off.  as they penetrate inland, no longer being supported with convection currents from  Additionally, as they move away from the tropics, TC's storm structure  weaken and they transition into extra-tropical systems with less organised convection and lower wind speeds, although still capable of continuing to bringing heavy precipitation if the conditions allow. This could partly explain why surge and wind, which rely on high wind speeds, affects less regions strongly south of QLD than flood and landslide hazards which rely on heavy and sustained precipitation.

An important consideration when evaluating flood and landslide hazards is that a cumulative method was used to calculate hazard values from input datasets. Rather than taking averages over each LGA as was done for surge and wind, flood and landslide input datasets were high-resolution raster maps with many null values. Using an averaging methodology would have described an LGA's hazard in proportion to its area, meaning larger LGAs with many flood-prone regions could still have a low flood hazard value. Instead, values were summed, meaning greater than zero hazard values meant a region had some hazard-prone regions, and high hazard values meant they had more regions prone to flooding/landslides regardless of the LGA's size. While this does mean larger LGAs have the potential to reach higher hazard values, this method represents all possible hazards, and therefore risk, rather than underestimating it due to averaging methods.

**4.4 TC Multi-hazard Risk**

Risk maps were created by multiplying each hazard with the exposure and vulnerability indices. This produced the four hazard-specific risk maps in Figure 10, from which a total TC risk map was created by equally weighting them as seen in Figure 11. A Natural Breaks symbology was used for these risk maps to group similar values and maximise variance between groups for a visually informative map.

[Figure]

[Figure]

[Figure]

**Figure 10~8~.** Risk maps for each hazard (surge, flood, wind, landslide).

[Figure]

**Figure 11~9~.** Combined multi-hazard risk calculated by equally weighting four hazard-specific risk maps.

Surge risk is considerable along the northern, western, and eastern coasts, with the highest values between Brisbane and Cairns in QLD. Flood risk can be seen to be highest across both NSW and QLD with medium values along the top of NT and WA. Risk to ~the~ wind is very uncommon at distances greater than 500 km inland and south of NSW, with the highest wind risk found along with the eastern parts of NSW and QLD. Landslide risk also shows the highest risk in eastern NSW and QLD with medium risk across northern NT. The combined TC risk map displays some of these more prominent patterns from each hazard-specific risk map. For example, eastern NSW and QLD have the highest risk followed by medium risk across northern WA and NT. The risk to TCs is very low inland of the country surrounding SA, as well as south of NSW, in VIC ~and Tasmania (TAS) states~ and TAS.

As patterns seen in risk maps can be partially explained by similar patterns found in constituent layers, it is important to compare them to hazard, exposure and vulnerability layers. While an overall

TC risk map is useful for such discussions, hazard-specific risks are important to consider and compare at a local level, for example when LGA councils are planning disaster management
550 strategies or communicating warnings to residents for an incoming TC.

Of note is that although TCs generally form over tropical  waters and affect regions near the tropics, they intensify away from the equator reaching maximum intensity approximately at 17-18°S of the equator (Kuleshov, 2020), which partially explains why risk is not highest in all northernmost LGAs. Another contributing factor to medium risk values in the country's north is due to there being
555 relatively fewer assets exposed compared to the rest of the country, as shown by the exposure index in Figure 6. Continuing moving further south away from the tropics, TCs are weakening as sea surface temperatures get colder in extra-tropical regions. Hence, substantial reduction of risk is observed in VIC and TAS. Similarly, TCs weaken over land which is why risk is also very low for central Australian LGAs. The lower risk in these states is supported by historical records of TC tracks from
560 1970-present (Kuleshov, 2020)

565  ~~Of note is that although TCs generally form over tropical waters and affect regions near the tropics, they intensify away from the equator reaching maximum intensity approximately 17-18°S of the equator (Kuleshov, 2020), which partially explains why risk is not highest in all northernmost LGAs. Another contributing factor to medium risk values in the country's north is due to there being relatively fewer assets exposed compared to the rest of the country, as shown by the~~
570 ~~exposure index in Figure 4. Continuing moving further south away from the tropics, TCs are weakening as sea surface temperatures get colder in extra-tropical regions. Hence, substantial reduction of risk is observed in VIC and TAS. Similarly, TCs weaken over land which is why risk is also very low for central Australian LGAs. The lower risk in these states is supported by historical records of TC tracks from 1970-present (Kuleshov, 2020).~~ The risk maps in Figures 10 and 11 attempt to
575 compare the relative risk of each LGA in Australia by summarising values of relevant hazard, exposure, and vulnerability indicators. Thus, we can investigate the risk in a certain region, and trace it back to its components' trends which should show a similar story. The highest risk regions identified along the eastern half of NSW and QLD in all risk maps (exclusively along the coast for surge risk) represents the dense  distribution of populations and infrastructure along
580 Australia's eastern states as seen in exposure maps in Figures 5 and 6. Accompanied by very high flood and landslide values, with wind and storm surge weakening in the southern half of NSW, the eastern strip of Australia stands out to have the highest risk. The influence of vulnerability has a less noticeable trend as it does not uniformly compound in all regions with high hazard and exposure, but can be seen to increase risk particularly in north-central NT, northern QLD and northern NSW.

585 This holistic approach to assessing risk is helpful in understanding the possible impacts if a TC was  to occur and affect any region in the country. Results have shown that this methodology is effective in visually describing and identifying regions with high risk component values, and hopes to provide relevant risk information to assist disaster management and resilience decision makers.

590 2.5. 4.5 Limitations of Risk Assessment

[revised manuscript text omitted]

Scawthorn, C., Schneider, P. J., and Schauer, B. A.: Natural hazards—The multihazard approach, 725 Natural Hazards Review, 39, 10.1061/(ASCE)1527-6988(2006)7:2(39),

Vintila, P.: Moving out: Aged migration in Western Australia 1991–96, Urban Policy and Research, 19, 203-225, 10.1080/08111140108727872, 2001.

Tropical Cyclones, last access: 12 March 2021.

Zhang, K., Xiao, C., and Shen, J.: Comparison of the CEST and SLOSH Models for Storm Surge 730 Flooding, Journal of Coastal Research, 242, 489-499, 10.2112/06-0709.1, 2008.

**5.  Appendix**

**Appendix 1**. Data table for LGA risk analysis. Links are provided for the data sources as well as the year that the dataset was last updated.

735

| Indicator | Dataset used | Source | Year |
|---|---|---|---|
| **Hazard** | | | |
| Surge hazard | Point feature layer of Storm surge run-up height, 100yr return period | GAR Atlas | 2015 |
| Flood hazard | Raster Flood depth inundation, 100yr return period | GAR Atlas | 2015 |
| Wind hazard | Raster Cyclone wind, 100yr return period | Geosciences Australia | 2018 |
| Landslide hazard | Raster Global landslides hazard | ARUP | 2020 |
| **LGA Exposure** | | | |
| Population density | Recorded total number of people living in each LGA. | ABS Census data | 2016 |
| Public hospital | Point feature layer of public hospitals around Australia | ArcGIS Online Dataset | 2019 |
| Substations | Point feature layer of power substations around Australia | Geosciences Australia | 2016 |
| Powerlines | Line feature layer of powerlines around Australia | Geosciences Australia | 2016 |
| **LGA Vulnerability** | | | |
| IRSD | Summary statistic for socioeconomic status, | ABS Census data | 2016 |
| No vehicle homes | Percentage of households within each LGA that owns zero vehicles. | ABS Census data | 2016 |
| Vulnerable age groups | Percentage of LGA population that is under 15 or over 65 | ABS Census data | 2016 |
| **Shape layers** | | | |

| LGA polygon layer | Shapefile containing the size of each LGA as of 2016 | ABS | 2016 |
|---|---|---|---|

---

## Author Comment (AC2)

**Document addressing reviewer comments**

- The manuscript provides a framework for comparing TC-related risk across Australia, incorporating multiple hazards, multiple exposure elements and multiple indicators of vulnerability.
- The framework described is a relative risk rating, based on a limited view of the probability of events (i.e. a 1% AEP level of hazard) in combination with national-scale indicators of exposure and vulnerability.
- Similar efforts have been undertaken within government in recent times, but are as yet unpublished. This manuscript provides a stimulating discussion on the complexity of evaluating multi-hazard risk in a nationally-consistent framework.
  - Properly undertaken, the resulting information from this analysis could be valuable for prioritising interventions across the country.
- The derivation of some metrics warrants further discussion the range of spatial scales presents unique challenges to developing representative rankings of hazards, especially with relatively coarse information. Flood and storm surge inundation are highly sensitive to spatial resolution, and will be challenging to represent at LGA resolution.
  - The elements of exposure and vulnerability must be linked using social vulnerability indicators in combination with physical asset exposure will not produce a valid evaluation of risk (either physical or social).

Dear Reviewer, thank you for your valuable comments which helped us to improve quality of the manuscript. All your comments have been addressed in a revised version of the manuscript. We hope you will find this revision satisfactory.

- 25 This is the ideal, having multiple vulnerability indicators specific to each exposure indicator, as well as tailored to each specific hazard, but due to limitations in data availability, a more general estimated approach was taken to estimate exposure as the combination of populations and built infrastructure, with vulnerability indicators of IRSD, vulnerable age groups etc. referring to that general exposure. Infrastructural vulnerability indicators were sought, but data not found to be available. We have added discussion of this limitation into the revised manuscript at the end of section 3.1.
  - We have concerns over the evaluation of risk, showing highest risk in northern NSW, including some LGAs well inland where wind hazard will be declining, there is no surge hazard and flood hazard is evaluated to be in the lowest quintile.
- At first this was thought to be due to high exposure values, but upon further inspection, hazard values in this high risk area actually have significant index values of ~0.4-0.7 despite original hazard maps suggesting they are in the lowest natural breaks class of hazard. As a result, hazard maps (as well as exposure and vulnerability indicator maps) have had a symbology change from natural breaks to quantile (decile for hazard and quintile for exposure/vulnerability). This depicted representative trends that are transferred over into risk calculations, and shows hazard to actually be quite high for flood in particular over NSW/lower QLD. This stark difference was because hazard values had not been normalised in the map to keep information for surge and wind (surge

5

10

15

|    | height and wind speed), however using natural breaks for data with values with very                      |
|----|----------------------------------------------------------------------------------------------------------|
| 45 | different magnitudes at min and max boundaries did not align with processed values that                  |
|    | were used in risk index calculation (decile normalised values). In the revised manuscript,               |
|    | maps have been updated to a quantile symbology, and for hazard, descriptive values have                  |
|    | been kept, but now the e.g. 2 nd highest decile (2nd darkest blue) is equivalent to 0.8 while |
|    | highest/darkest is equivalent to 0.9 in the risk calculations.                                           |

- Sections Method, Results and Discussion have been updated to reflect these changes. The Quantile symbology has been used for exposure and vulnerability indicators as well, however the trends/findings from the maps have not changed, as indicator data had already been transformed into a decile format.

**55 Specific comments and questions for the authors:**

• Line 78: Only one ARI is used - the relative impact may change for different return levels due to the different spatial pattern in hazard and/or vulnerability. Do the authors have any comments on this?

One ARI return period was used due to data availability. Although some hazard datasets had more ARI periods available, 100 years was chosen as it was available for all chosen hazards.

• Line 79: It is not appropriate to say this is representative of the hazards in the near future. The 100-year ARI hazard level is an indication of the long term probability in that it occurs - on average - once in every 100 years. There is a possibility of such an event occurring in any given year (approximately a 1% probability), with no inference about the near future likelihood. Further, the 100-year return period level may well change over the next 100 years. Knutson et al (2020) report the most confident TC-related projection is increased storm surge levels, with medium to high confidence that TCrelated precipitation will increase at the global scale.

**This has been revised, removing 'in the near future' and adding that it is the 100-year ARI level at the moment and is predicted to change in the future.**

• Section 2.1: There is not sufficient discussion on the metrics used for the hazard indices. No references are provided for flood or landslide hazard information in the main part of the manuscript (a table is presented in the appendix, but it is not referenced, and the links in the table are not accessible); the reference provided for Storm Surge does not describe that hazard ("For this global study, the effects are only related to the wind speed at a global scale." Cardona et al., 2014). This is a major concern to the core objectives of the manuscript.

The data table has been moved to the main part of the manuscript (as per another reviewer's comments), and has now been referenced. Upon further inspection of information attached to the surge data taken from GAR Atlas' risk data platform, Cardona and Bernal references do not seem to align, and at the moment it is unsure which company/researchers created it. As a result, GAR Atlas has been referenced, and a more detailed description of the surge data is included under the methods hazard section, along with more descriptions of wind/landslide/flood input data.

 Line 80: There are more up-to-date sources of information for storm tide hazard - e.g. the Canute 3.0 data available through the NESP Climate Hub (https://shiny.csiro.au/Canute3\_0/)

65

75

85

We were not aware this existed at the time of conducting the study, and the surge data from GAR Atlas fit our methods and was easy to summarise to LGA level. More up to date data will be considered for future work trying to improve upon the hazard index.

Line 81: Mean values of hazard may not be appropriate for some LGAs. This is an issue the authors note (in reference to East Pilbara). However, the hazard needs to be evaluated in the context of exposed assets. In the case of East Pilbara, the majority of exposed assets (primarily population) are close to the coastline, where wind hazard (and flood hazard) will be higher. In our comparative rankings, we have used a 90th percentile of the hazard level, reflecting the general proximity of population to the coastline.

This is a limitation that is discussed, however the approach we took was to apply an easily replicable methodology, using the same processes and calculations for all LGAs, rather than personally identifying outliers, and applying our own assumptions to them. This is because
interventions would require a strong understanding of all regions in Australia, and we would need to choose a cut off point for where we do not intervene. In this case, ideally a higher resolution would reduce the occurrences of large LGAs disguising the exact locations of high/low exposure such as in East Pilbara.

- Line 89: No reference to the table of data sources is provided.
- 105 References to data sources from past studies have been added in the table and reference list, and hyperlinks have been used for ABS/webpage sources.
  - Line 89: Power line and electrical substations will be highly correlated, so using both as input to the exposure definition will be unduly weighted to that infrastructure element.
- Correlations were also found between population and hospital density along with powerline and
   substation datasets. This is because these types of infrastructure are built to meet the demand of
   populations living there. It is true there are 3 infrastructure indicators vs one population indicator,
   but all are chosen as important and valuable assets, equally weighted in the exposure index
   calculation. As the trends in each dataset are very similar, removing one of powerline/substation
   indicators would not make a substantial difference to the index.
- Line 89: What power line information was used distribution lines or transmission lines? In some urban LGAs, there may be limited transmission network coverage, with power supplied through lower voltage feeder networks that may lead to biased estimates of exposure. The data table provided does not contain working links, so readers are not able to inspect those sources.
- 120 Transmission powerlines of high voltage electricity were used. The link should be provided this time. While yes, some LGAs where the larger transmission powerlines will not pass through will show lower exposure, transmission powerlines represent the powerlines we determined to be more valuable/critical, as dysfunction would power out the subsequent smaller distribution lines.
- 125
- Overall losses will be impacted by the value of lost income to businesses. With no business information included, this may lead to an underestimate of exposure in some areas.

This is exactly correct, and discussed in relation to the mining economy in WA, as well as in limitations of risk assessments being limited to the information of chosen indicators. In this case,

**we prioritized human life and infrastructure, whereas another study for end users that areinterested in GDP and productivity of different industries would use relevant exposure indicators.**

• Line 103: The choice of vulnerability indexes is not clearly linked to the choice of exposure indexes. In the Hazard-Exposure-Vulnerability framework, the vulnerability is directly related to the exposed assets. Using social vulnerability indicators and physical assets presents a logical mismatch between the two risk factors. Ideally, physical vulnerability indicators should be used that link the hazard to the exposed physical assets.

Roughly the same comment was made above, and has been addressed there. To summarise, we have noted that ideally exposure and vulnerability indicators are directly linked (social vulnerability -> populations, building code -> infrastructure), however due to limitations in data, this was not feasible.

- The "no vehicle homes" is duplicated in the contributing indicators in the IRSD indicators, so places undue weighting on this indicator. Additionally, the claim of "no vehicle homes" indicator being particularly relevant should be justified what evidence is there to support the assertion they are more susceptible to loss of life, especially given the very limited fatalities attributable to TCs in Australia? Further, evacuation is only a consideration in storm tide prone areas. Otherwise, the emergency services advice is to shelter in place (i.e. at dwellings that are built to modern codes). A better indicator of vulnerability would therefore be the proportion of houses that are not constructed to modern wind loading standards.
- 150 The possible duplication of no vehicle homes indicator with 'NOCAR' input in IRSD is discussed in the vulnerability discussion already. The point about an evacuation indicator for a historically non-lethal natural disaster to Australia is a good point, and we would prefer to have an indicator that tells us the vulnerability of the infrastructure that cannot be evacuated and historically is a big portion of the damages/loss.
- 155 This suggestion is incorporated stating that "Although this risk assessment highly values human life and safety, historically within Australia, TCs have caused very few fatalities in recent decades, and an indicator describing the vulnerability of infrastructure would be preferred. An alternative to 'no vehicle homes' vulnerability indicator could be the proportion of houses that are not constructed to modern wind loading standards. While this potentially useful indicator was not included in this study due to limited data availability, this could be a topic for future work. "
  - Line 274: The use of data with null values for some LGAs suggests additional effort is required to ensure consistent coverage either through alternate indexes or suitable estimations from other sources.
- 165 LGAs with null values were generally less populated/country LGAs, and were very uncommon (mentioned LGA of Maralinga Tjarutja was one of the only cases, with no data values for IRSD data from the ABS).
  - Line 307: Correct "main coastland"

Updated to 'mainland'.

135

140

|     | Cameron Do and Yuriy Kuleshov                                                                                                                                                                                                                                                                                             |
|-----|---------------------------------------------------------------------------------------------------------------------------------------------------------------------------------------------------------------------------------------------------------------------------------------------------------------------------|
|     | Multi-hazard Tropical Cyclone Risk Assessment for Australia                                                                                                                                                                                                                                                               |
| 185 |                                                                                                                                                                                                                                                                                                                           |
|     | The hyperlinks work on the word document we submitted – we will make a note to the publishers to ensure links are working in the next iteration of the manuscript.                                                                                                                                                        |
| 180 | • Appendix: None of the links in the table are accessible - appears the links have not been properly included in conversion to PDF. "Geosciences Australia" should be "Geoscience Australia"                                                                                                                              |
|     | Scawthorn has had as many details as possible available added to it. Do and Kuleshov reference has been removed for a flood risk assessment reference (Amadio et al., 2019) rather than an Australian TC context, as the paper is still undergoing its publishing/acceptance process. Burston reference has been updated. |
| 175 | • Several of the references are incomplete or inaccessible e.g. Scawthorn et al., 2006, Do and Kuleshov, 2022, Burston et al. (missing journal name)                                                                                                                                                                      |
|     | Included a sentence specifying importance for hazard indicators as exposure/vulnerability indicators from the ABS are well validated.                                                                                                                                                                                     |
| 170 | • Line 377: ABS data would typically be well validated. Engagement with the ABS may have addressed the authors concerns over validation of the (vulnerability) indicators.                                                                                                                                                |

Natural Hazards and Earth System Sciences https://www.natural-hazards-and-earth-systemsciences.net/

**Abstract**

Tropical cyclones (TCs) have long posed a significant threat to Australia's population, infrastructure, and environment. This threat may grow under climate change as projections indicate continuing sea level rise and increases in rainfall during TC events. Previous TC risk reduction-assessment\_efforts 195 have focused on the risk from wind, whereas a holistic approach requires multi-hazard risk assessments that also consider impacts of other TC-related hazards. This study assessed and mapped TC risk nationwide, focusing on the impacts on population and infrastructure from the TC-related hazards of wind, storm surge, flooding and landslides. Risk maps were created at the Local Government Area (LGA) level for all of Australia, using collated data on multiple hazards, exposure 200 and vulnerability. The results study-demonstrated that the risk posed by all hazards was highest for coastal LGAs of eastern Queensland and New South Wales followed by medium risk across Northern Territory and north-west of Western Australia., with flood and landslide hazards also affecting several inland LGAs. Further enhancement improvement and validation of risk maps developed in this study will provide decision-makers with the information needed to reduce TC risk, save lives and 205 prevent damage to infrastructure. The resulting maps of risk will provide decision-makers with the information needed to further reduce TC risk, save lives, protect the environment, and reduce economic losses

Comment [CD1]: Reviewer: acronym table

5

| Acronyms     |                                  |
|--------------|----------------------------------|
| ABS          | Australian Bureau of Statistics  |
| LGA          | Local Government Area            |
| NSW          | New South Wales                  |
| NT           | Northern Territory               |
| QLD          | Queensland                       |
| SEIFA | Socio-Economic Indexes For Areas |
| SA    | South Australia                  |
| TAS          | Tasmania                         |
| TC           | Tropical Cyclone                 |
| VIC          | Victoria                         |
| WA           | Western Australia                |

**210 **1. Introduction**

215

Tropical Cyclones (TCs), also known as hurricanes or typhoons, are powerful and highly destructive meteorological hazards. Since 1970, almost 2,000 natural disasters have been attributed to TCs, which has led to over 700,000 deaths worldwide (World Meteorological Organisation, 2021). Costing about U.S.\$26 billion annually in global damages (Mendelsohn et al., 2012), their impact is expected to multiply to U.S.\$60 billion annually by 2100 (Bakkensen and Mendelsohn, 2019).

The proportion of intense TCs (Saffir–Simpson scale categories 4-5 with >209km/h-1-minute maximum sustained winds >209km/h) and peak wind speeds of the most intense TCs are projected to increase at the global scale with increasing impact of global warming (high confidence) (IPCC, 2022)-(IPCC AR6) (IPCC, 20221). The potential of more destructive TC events will require updating

220 and enhancement of existing risk reduction strategy. The Sendai Framework for Disaster Risk Reduction provides a structure for reducing disaster damages and increasing resilience to hazards including TCs (Bennett, 2020). One mechanism they encourage in Goals 18 and 24 is the distribution of multi-hazard risk information such as risk assessments.

Risk assessments combine hazard information with human activity, infrastructure and natural

- 225 resources to determine the possible impacts of hazardous events (Belluck et al., 2006; National Research Council, 1991) and make informed choices for risk management in the most exposed and vulnerable regions (Aguirre-Ayerbe et al., 2018). Disaster risk is defined as the probability of harmful consequences, or significant losses, resulting from interactions between a hazard, and the local exposure and vulnerability to that hazard (Crichton, 1999; Downing, 2001).
- 230 As Local Government Areas (LGAs) are the one of the smallest government decision-making bodies with available census datay, information is sought to be provided on that scale. Risk assessments are a foundation for early warning systems to raise alerts of potential impacts, and to provide evidence for the prioritisation of funds and resources to areas in advance of any hazardous events. While the climate continues to change alongside evolving human activity, risk assessments must likewise be
- regularly updated to stay accurate and useful as a tool for disaster risk reduction (Peduzzi et al., 2012).

For TCs, the four main hazards are the destructive winds, associated storm surge, flooding from associated heavy rainfall, and landslides on steep terrain as soils saturate (Murray et al., 2020). TCs and other natural hazards are becoming increasingly recognised as multi-hazardous in nature (Scawthorn et al., 2006)-(Scawthorn et al., 2006a). These hazards impact regions differently and their effects can compound to cause even greater damage (Gori et al., 2020).

While TCs can cause damage through different hazards, such as gale-force winds, storm surge or flooding, the communication of TC intensity and categorisation places emphasis on wind speed (Lavender and Mcbride, 2020). This is partially due to the availability of wind measuring technology and the relative ease to quantify wind. Publicly available warnings and forecasts are focusing on wind speeds, ultimately portraying the message that winds are the hazard to be most wary of. The literature however suggests the TC-induced impacts of storm surge and flooding contribute to the most human lives lost and infrastructure damage (Mendelsohn et al., 2012; Zhang et al., 2008).

Although some studies have included multi-hazard aspects of TCs\_(Burston et al., 2017) (Burston et al., 2017b),-presenting different hazard models for TC-induced storm surge, wind and flooding, these studies do not complete the story of combining hazard with exposure and vulnerability to map risk. Similarly, within the literature, there are many examples of standalone exposure or vulnerability index assessments for TCs (Marín-Monroy et al., 2020; Bathi and Das, 2016; Amadio et al., 2019). This gap indicates compelling scope to develop a multi-hazard TC risk assessment that can

255 differentiate the extent and severity of TC-<del>related</del>-induced hazards.

This study will address this gap and strengthen TC risk information for the Australian region. Multihazard risk is assessed and visualised through-interactive maps which show LGA categorisation, alongside hazard, exposure, and vulnerability layers. As a risk assessment's usefulness relies on how they are tailored for <del>a</del> specific users audience or applications, the method proposed in this study serves as a proof of concept that can be altered in future iterations for tailored use.-

**Figure 1**. Map of study area, state, and territory boundaries as well as Local Government Area (LGA) divisions and major cities. States and territories: Western Australia (WA), Northern

numbering

7

240

245

**265 Territory (NT), Queensland (QLD), South Australia (SA), New South Wales (NSW), Australian Capital Territory (ACT), Victoria (VIC) and Tasmania (TAS) are labelled.**

Table 1. Comparison table of each Australian states general characteristics including total area, real GSP and population (Australian Bureau of Statistics, 2020-2021)

**Formatted Table**

| STATE                 | TOTAL AREA    | Number         | Avg. area per   | GSP        | Population     | GSP per      |
|-----------------------|---------------|----------------|-----------------|-------------------|----------------|--------------|
|                       |               | ot lgas | LGA             | (Şmillion) |                | capita (Ş)   |
| Australia Capital     | 2358          | 1       | 2358.172        | 433740            | 431483         | 100523       |
| Territory (ACT)       |               |                |                 |                   |                |              |
| New South Wales       | 800811 | 130     | 6160.083 | 6336350    | 8172561 | 77532        |
| (NSW)          |               |                |                 |                   |                |              |
| Northern Territory    | 1348094       | 18      | 74894.13        | 261810            | 246565         | 106183       |
| (NT)           |               |                |                 |                   |                |              |
| Queensland (QLD)      | 1730172       | 78      | 22181.69        | 3689770           | 5194884        | 71027        |
| South Australia       | 984275 | 71      | 13863.03 | 1149210           | 1770794        | 64898 |
| (SA)           |               |                |                 |                   |                |              |
| Tasmania (TAS) | 68018         | 29      | 2345.443        | 340830            | 541499         | 62942        |
| Victoria (VIC)        | 227496        | 80             | 2843.695        | 4682640           | 6661697        | 70292        |
| Western Australia     | 2526646       | 137     | 18442.67        | 3206530           | 2670231        | 120084       |
| (WA)           |               |                |                 |                   |                |              |

270

Australia is a country with a long coastline and with much of its northern states commonly impacted by tropical cyclonesTCs. An average of 12 TCs form in the Australian region annually (however, interannual variability is high ranging from 19 TCs in 1983/84 to 3 TCs in 2015/16, for records examined from 1970/71 to 2019/20 TC seasons (Kuleshov et al., 2020)), with 5 making landfall on average (Mortlock et al., 2018). In the last few decades, several severe TC events have destroyed 275 infrastructure and caused billions of dollars in losses, including TC Larry (2006), TC Yasi (2011) and TC Debbie (2017).

Figure 1 shows the boundaries of each state and territory as well as the outline of Local Government Area (LGA) divisions within. Table 1 summarises key traits of each state such as their total area, real Gross State Product (GSP) and population. From the Ttable 1 it can be seen that NSW and VIC are the states with the highest GSP (monetary measure of state output), as well as highest populations. For TC-related impacts however, we are most concerned with interested in the northern states that are expected to more commonly be impacted by TC events. QLD and WA therefore stand out as the next most important states with next highest GSP and populations. Important to note however is the size of QLD and WA states and much higher average area per LGA, meaning GSP contribution and populations are likely to be much more spread out.

**2.3. Data and Methodology**

To calculate the multi-hazard risk of TCs to Australia, hazard, exposure exposure, and vulnerability datasets were chosen and sourced. This data was then joined combined to LGA map shapefiles in ArcGIS Pro. To calculate exposure and vulnerability indexes from multiple indicators, equal weighting was used for exposure, while Pareto front-ranking was used for vulnerability. Combined with hazard values for each LGA, exposure and vulnerability indexes were used to calculate risk using equation 1:

290

280

Risk = Hazard x Exposure x Vulnerability

(1).

1.1.-3.1 Selection of indicators

| Table 2. Dat                | ta table for LGA risk analysis. Lin                                                               | ks are provided for the data                  | sources as well | as •                                            | Formatted: Normal, No bullets or numbering |
|-----------------------------|---------------------------------------------------------------------------------------------------|-----------------------------------------------|-----------------|-------------------------------------------------|--------------------------------------------|
| the year that the ua | llaset was last upualeu.                                                                          |                                               |                 |                                                 |                                            |
| Indicator                   | Dataset used                                                                                      | Source                                        | Year     | Data forma
and
resolution                 | t                                   |
| Hazard               |                                                                                                   |                                               |                 |                                                 |                                            |
| Surge hazard                | Global tropical cyclone storm
surge run-up height, 100yr
return period | GAR Atlas
(2015)                    | 2015            | Point data
(every 1km
along
coastline) | Field Code Changed                         |
| Flood hazard                | Australian flood depth                                                                            | GAR Atlas                                     | 2015            | Raster data                                     | Formatted: Not Highlight                   |
|                             | inundation, 100yr return                                                                          | (Rudari et al., 2015)                         |                 | (1km)                                    | Formatted: Not Highlight                   |
| Wind hazard                 | Australian tropical cyclone
wind, 100yr return period                                          | Geoscience Australia
(Arthur, 2018) | 2018     | Raster data
(2km)                     |                                            |
| Landslide hazard            | Global landslides hazard                                                                          | ThinkHazard!
(Arup, 2020)           | 2020            | Raster data
(1km)                     |                                            |
| LGA Exposure                |                                                                                                   |                                               |                 |                                                 |                                            |
| Population                  | Recorded total number of
people living in each LGA.                                     | ABS Census data                               | 2016     | Tabular data
(LGA
resolution)      | à                                   |
| Public hospital             | Point feature layer of public hospitals around Australia                                          | ArcGIS Online Dataset                         | 2019            | Point data                               |                                            |
| Substations          | Point feature layer of power
substations around Australia                                      | Geoscience Australia                          | 2016            | Point data                               |                                            |
| Powerlines           | Line feature layer of powerlines around Australia                                                 | Geoscience Australia                          | 2016            | Line data                                |                                            |
| LGA Vulnerability           |                                                                                                   |                                               |                 |                                                 |                                            |
| IRSD                        | Summary statistic for
socioeconomic status,                                             | ABS Census data                               | 2016     | Tabular data
(LGA
resolution)      | à                                   |

| No vehicle homes         | Percentage of households
within each LGA that owns
zero vehicles. | ABS Census data                                  | 2016 | Tabular data
(LGA
resolution)        |
|--------------------------|-------------------------------------------------------------------------|--------------------------------------------------|-------------|----------------------------------------------------------|
| Vulnerable age
groups | Percentage of LGA
population that is under 15
or over 65   | ABS Census data                                  | 2016 | Tabular data
(LGA
resolution)               |
| Shape layers             |                                                                         |                                                  |             |                                                          |
| LGA polygon layer        | Shapefile containing the size
of each LGA as of 2016                 | ABS Australian Statistical
Geography Standard | 2016 | Polygon
data (LGA
resolution) |

**Hazard**

The main identified hazards of TCs include storm surge, winds, landslides, and floods. The 100 year100-year return period was chosen to represent the current long-term probability-danger of these hazards occurring, in the near future. Of note is that these probabilities may change in the future with studies predicting increased storm surge levels (high confidence) and increased TCrelated precipitation (medium-high confidence) due to climate change (Cha et al., 2020).

305

315

325

300

Storm surge data was acquired from GAR Atlas' risk and data platform, which mapped TC storm surge height as point data roughly along the Australian coastline every 1km. TC wind data was sourced from Arthur (2018) and came as high-resolution raster data over Australia and its northern waters. Flood data was sourced from Rudari et al. (2015) as high-resolution raster data representing riverine flooding only. Thus, non-null values tended to only appear near riverine systems and catchments. Similarly, landslide data from Arup (2020) was in the raster format with mostly null 310 values apart from specific locations with significant landslide hazard.

[revised manuscript text omitted]

**Comment [CD2]:** Reviewer: move appendix up here - add columns on data format and resolution

the LGA resolution. As raw data came in tabular (a), point (b) and raster (c) formats, different methods for each were used to summarise information when converted to LGA polygons (d) as depicted in Figure 2. Tabular data from the ABS came at an LGA resolution, so data only needed to be linked to an LGA polygon shapefile in ArcGIS Pro. For storm surge point data which spanned across the coastline every 1km, the average 100-yr surge height value was taken, whereas for exposure point data such as hospitals and substations, the count or number of points in each LGA was taken. For the wind raster data which had no null values and gradually changed in value inland, the mean windspeed value was taken per LGA, while with flood and landslide data the sum of nonnull values was taken per LGA.

375

380

**Figure 2**. Diagram representing data formats of acquired raw data (tabular (a), point (b), raster (c)) being transformed into a comparable LGA polygon format (d). (Example data is used here, and (d) is not representative of any results)

Once in a comparable data format, indicator data values were normalised into a 0 to 1 range with decile normalisation against the whole country using Python scripts. Use of different normalisation methods were tested, such as linear normalisation and natural breaks, however decile normalisation was found to best remove the skewing effects of outliers, and is a method commonly used in several ABS indices.

Figure 3 depicts the different tiers or stages of the risk assessment, starting at tier (3) with the
 indicators. These are the variables that differ in value spatially across Australian LGAs, that were
 chosen to be representative of TC hazard, exposure, or vulnerability. Three to four indicators were
 chosen to give a more robust index without diluting the sensitivity of each indicator. From tier (3) to
 tier (2), or from indicators to indices, different methods were used depending on the index. For
 explained in the next-sub-section (3.3). With only one indicator or dataset for each hazard of TCs,
 each hazard was passed through separately, meaning when hazard, exposure, and vulnerability
 indices were multiplied as outlined inusing equation 1 to calculate risk (tier (2) to tier (1)), four

Comment [CD3]: Delete